



# Measurement report: Receptor modelling for source identification of urban fine and coarse particulate matter using hourly elemental composition

Magdalena Reizer[1], Giulia Calzolai[2], Katarzyna Maciejewska[1], José A. G. Orza[3], Luca Carraresi[2], Franco Lucarelli[2], Katarzyna Juda-Rezler[1]

[1] Faculty of Building Services, Hydro and Environmental Engineering, Warsaw University of Technology, Warsaw, Poland
[2] Department of Physics and Astronomy, University of Florence and National Institute of Nuclear Physics (INFN), Sesto Fiorentino, Italy
[3] SCOLAb, Física Aplicada, Universidad Miguel Hernández, Elche, Spain

*Correspondence to*: Magdalena Reizer (magdalena.reizer@pw.edu.pl)

**Abstract.** Elemental composition of the fine ($PM_{2.5}$) and coarse ($PM_{2.5-10}$) fraction of atmospheric particulate matter was measured at hourly time resolution by the use of a "streaker" sampler during a winter period at a Central European urban background site in Warsaw, Poland. A combination of multivariate (Positive Matrix Factorization), wind- (Conditional Probability Function) and trajectory-based (Cluster Analysis) receptor models, was applied for source apportionment. It allowed for identification of 5 similar sources in both fractions, including sulfates, soil dust, road salt, traffic- and industry-related sources. Another 2 sources, i.e., Cl-rich and wood and waste combustion, were identified in the fine fraction solely. In the fine fraction, aged sulfate aerosol related with emissions from solid fuel combustion in the residential sector located outside the city was the largest contributing source to fine elemental mass (44%), while traffic-related sources, including soil dust mixed with road dust, road dust, as well as exhaust and non-exhaust traffic emissions, had the biggest contribution in the coarse elemental mass (together accounting for 83%). Regional transport of aged aerosols and more local impact of the rest of identified sources played a crucial role in aerosol formation over the city. In addition, 2 intensive Saharan dust outbreaks were registered on 18th February and 8th March 2016. Both episodes were characterized by long-range transport of dust at 1 500 m and 3 000 m over Warsaw, as well as the concentrations of the soil component being 7 (up to 3.5 µg m⁻³) and 6 (up to 6.1 µg m⁻³) times higher than the mean concentrations observed during non-episodes days (0.5 µg m⁻³ and 1.1 µg m⁻³) in the fine and coarse fraction, respectively. The set of receptor models applied to the high time resolution data allowed to follow in detail the daily evolution of the aerosol elemental composition and to identify distinct sources contributing to the concentrations of different PM fractions, as well as revealed "multi-faces" of some elements, having diverse origin in the fine and coarse fraction. The hourly resolution of meteorological conditions and air mass back trajectories empower to follow transport pathways of the aerosol as well.



## 1 Introduction

Nowadays, the exposure to both ambient and indoor air pollution has been identified as the biggest environmental risk to human health, with particulate matter (PM) being most commonly used as a proxy indicator of exposure to air pollution in general (WHO, 2016). Many epidemiological studies have shown strong relationship between PM and adverse health effects, focusing on either short-term or long-term exposure (e.g., Pope and Dockery, 2006; Pope et al., 2018). The greatest risk to health is posed by $PM_{2.5}$ (particles with aerodynamic diameter smaller than 2.5 µm), as it can penetrate the respiratory system via inhalation, causing or aggravating respiratory and cardiovascular diseases, reproductive and central nervous system dysfunctions, as well as cancer (e.g., Manisalidis et al., 2020). Globally, ambient $PM_{2.5}$ air pollution contributed to 4.14 million deaths in 2019 (Murray et al., 2020). In addition, the World Health Organization (WHO) estimates that about 90% of people living in cities are exposed to $PM_{2.5}$ levels exceeding the WHO annual guideline value of 10 µg m$^{-3}$ (WHO, 2018). According to the future projections, by 2040, effective implementation of the current policies would reduce the global anthropogenic primary $PM_{2.5}$ emission by about 10%, however global population-weighted annual mean $PM_{2.5}$ concentrations would increase by 10% (Amann et al., 2020).

Having this risk at mind, the identification of sources is needed, being one of the most crucial issues for development of air quality improvement strategies and the design of Air Quality Plans and programs (e.g., Viaene et al., 2016). Particularly in urban areas the problem of elevated PM concentrations is highly complex due to simultaneous occurrence of multiple sources of primary PM emissions and of formation processes of secondary particles. In order to identify and apportion ambient concentrations to sources of PM, different multivariate receptor models (RMs), ranging from simple techniques applying elementary mathematical calculations and basic physical assumptions, up to complex models requiring pre- and post-processing of data are commonly used (e.g., Belis et al., 2013; Hopke et al., 2020). These techniques are often complemented by the use of wind- and trajectory-based RMs, utilizing wind speed/direction measured at the receptor site and backward trajectories generated with a Lagrangian model, respectively (e.g., Belis et al., 2013). A number of studies dedicated to PM source apportionment have used 24-hour averaged concentrations. In Europe, the choice of this time step derives mainly from the need to align the sampling set up with the reference method for the determination of $PM_{2.5}$ and $PM_{10}$ mass, as well as to accomplish full chemical characterization of PM with collection of proper amount of material, in particular for quantifying PM components that are present in very low concentrations (e.g., Belis et al., 2019).

However, daily samples are not able to capture the dynamics of most of the emission processes and chemical reactions which PM undergoes in the atmosphere within a few hours. Sampling with higher time resolution, i.e., 1 hour, might improve identification of many PM sources, in particular those characterized by clear diurnal variations with peak concentrations during specific time of the day, e.g., traffic emissions or biomass/wood burning, as well as it can provide information on the processes of the built-up of PM episodes. In addition, hourly time resolution supplies 168 samples per week, thus even short-term sampling campaigns provide a reasonable number of samples for RMs, which have to be gathered in order to sufficiently resolve the PM sources (Hopke et al., 2020).




Over the last 10 years, a majority of studies using RMs have been applied to the PM samples in lower temporal resolution, typically 12-24 h (e.g., Belis et al., 2013; Hopke et al., 2020). Only limited number of studies have applied receptor
modelling techniques to the high-time resolution PM samples, particularly applying Positive Matrix Factorization (PMF) model to the hourly elemental composition of PM$_{2.5}$ and PM$_{2.5-10}$ samples. Identification of PM sources based on aerosol samples in 1-hour resolution has been carried out in Southern Europe, e.g.: at 4 urban sites in Barcelona (Spain), Porto (Portugal), Athens (Greece) and Florence (Italy) (Lucarelli et al., 2015); at an urban site in Elche (Spain) (Nicolás et al., 2020), at 6 sites of different types in Tuscany (Central Italy) (Nava et al., 2015) and in an industrial area of Taranto (Italy)
(Lucarelli et al., 2020). Outside this region, hourly-resolved PM samples have been investigated e.g.: in London (Crilley et al., 2017) and Port Talbot, South Wales (Taiwo et al., 2014a) in the United Kingdom, in Pearl River Delta region in China (Zhou et al., 2018), as well as at 4 different sites in Alexandra in New Zealand (Ancelet et al., 2014). However, according to our knowledge, receptor modelling studies based on hourly elemental composition of PM has not been carried out in Central Europe previously.
The aim of this study is source apportionment of urban fine (PM$_{2.5}$) and coarse (PM$_{2.5-10}$) fractions of atmospheric particulate matter with 1-hour time resolution. To this end, Positive Matrix Factorization, complemented by the wind- and trajectory-based receptor models, i.e., Conditional Probability Function and clustering of air mass back trajectories, respectively, has been applied to the elemental composition of both fractions. The measurement campaign has been carried out in Warsaw, Poland, during winter period of 18$^{th}$ February – 10$^{th}$ March 2016. The quantitative information on a broad range of elements
allows for a detailed characterization of the composition of urban aerosol. That makes it all the more important that as was shown by Juda-Rezler et al., 2021, some of the elements are highly bioavailable and their bioavailability increases during episodes of PM pollution. In addition, such high temporal resolution of source apportionment leads to the detailed identification of PM sources together with their possible area of origin and provide unique information for the implementation of effective PM mitigation strategies in the investigated urban area, in particular during wintertime.
Moreover, this study supplements the previous SA analysis of a 1-year long measurement campaign of concentrations of PM$_{2.5}$ and its main constituents, carried out in the same area (Juda-Rezler et al., 2020).

## 2 Materials and methods

### 2.1 Study area

The measurement campaign was conducted in Warsaw, the biggest city in Poland, with almost 2 million inhabitants and 3
462 persons km$^{-2}$ population density (as of 2019). It is located in central-east part of the country in the lowlands (78 – 112 m a.s.l.) along the banks of the Vistula River, which splits the city in the N-S direction and constitutes the most important ventilation corridor. Warsaw's total area amounting to 517 km$^2$ is in 22% covered by green areas. In addition, to the NW a 385-km$^2$ Kampinos National Park is located, where forests account for around 70% of the area. The studied area is presented in Fig. 1.




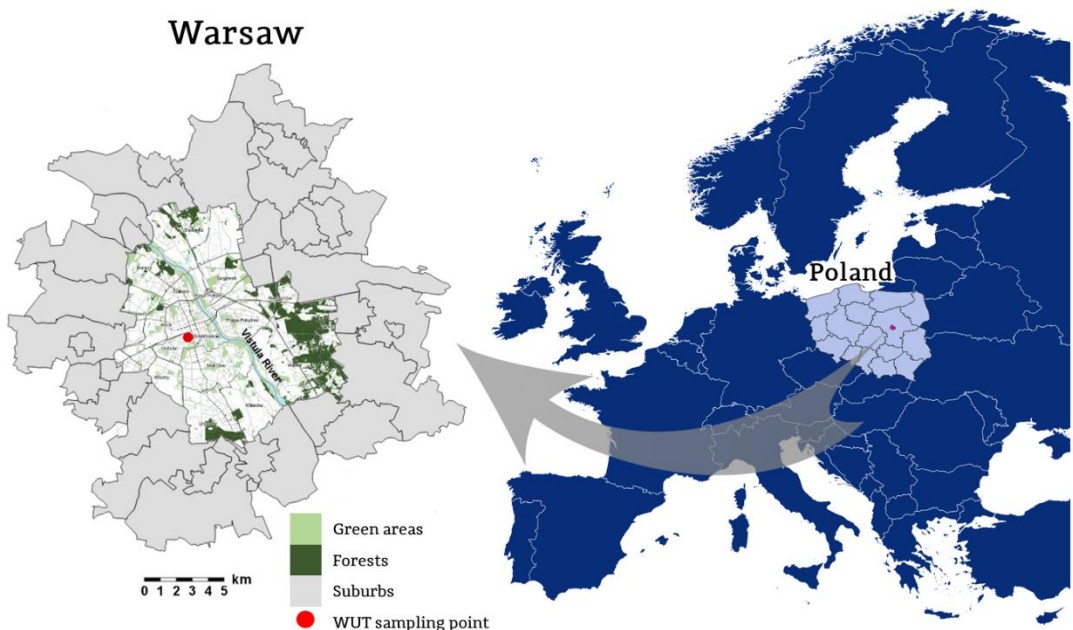

**Figure 1: Location of measurement site in Warsaw.**

Warsaw is characterized by moderately warm climate, with annual mean temperature of 8.5 °C and annual sum of precipitation around 530 mm (data for 1986 – 2010)[1]. January, with average temperature of −1.9 °C and July, with average temperature of 19.0 °C are the coolest and the warmest months, respectively. July is also characterized by the highest precipitation (72.9 mm), while in February the lowest precipitation is observed (26.1 mm).

Ambient air quality in the city is determined primarily by road transport emissions. Warsaw is characterized by a dense road network with the total length of more than 2 800 km, as well as the highest number of cars per 1000 inhabitants in the country equalling 750. In the same time, frequent congestions are also observed in the city. The municipal economy is not concentrated on industrial production, however, several point emission sources are located within the city, including two coal-fired combined heat and power plants: Siekierki (heat capacity 2 078 MW$_{th}$, power generation capacity 620 MW$_e$) and Żerań (1 280 MW$_{th}$, 373 MW$_e$), as well as a steel plant with annual production of 600 000 tons of steel products. Warsaw has the largest central heating supply system in the European Union, with over 1 800 km network, covering roughly 80% of the city's heat demand. However, the city is surrounded by numerous smaller suburb towns dominated by individual household heating, mainly using solid fuels (coal, wood) and gas, but also solid fuels of very low-quality, used engine oil and wastes (see e.g., Juda-Rezler et al., 2021).

In order to choose the location of the sampling site, representative for the urban background concentrations, air pollution dispersion modelling by the CALMET/CALPUFF system has been conducted (see Maciejewska, 2020). The site of Warsaw

---

[1] All meteorological data comes from the Institute of Meteorology and Water Management – National Research Institute, on-line access on 08/08/2020.



University of Technology (WUT) was placed in the city center within a 34-hectare area of the city's water treatment station and was beyond the direct influence of any particular local emission source or traffic.

## 2.2 PM sampling and elements determination

The sampling campaign was carried out during winter, between 18th February and 10th March 2016. The aerosol was collected by a sampling devices (PIXE International Corporation (Calzolai et al., 2015)) designed to separate the fine (<2.5 μm aerodynamic diameter) and the coarse (2.5–10 μm) modes of atmospheric aerosol. Coarse particles are collected on an impaction surface made up of an Apiezon-coated polypropylene foil, whereas fine particles are collected on a Nuclepore filter. The two collecting subtrata are paired on a cartridge that rotates under the air inlet at constant speed for a week. Thus, on each one of the two stages, a circular continuous deposition of particulate matter (the "streak") is formed; streaks are analyzed by means of the PIXE (Particle Induced X-Ray Emission) technique using a scan system. The proton beam size for PIXE analysis, together with the pumping orifice width and the cartridge rotation speed, determine the time resolution on the elemental composition of PM, which is one hour.

PIXE measurements were performed at the INFN-LABEC laboratory in Florence with a 2.7 MeV proton beams extracted from the 3 MV HVEE Tandetron accelerator. The external beam set-up, fully dedicated to environmental analysis, is extensively described elsewhere (Lucarelli, 2020). As aforementioned, the deposit streak was analyzed "point by point" with steps matching 1 h of sampling. Each point was irradiated for about 60 s with a 50–300 nA beam current. PIXE spectra were fitted by means of the GUPIX code (Campbell et al., 2010) and elemental concentrations were obtained by a calibration curve from a set of thin standards of certified areal density (Micro matter Inc., Surrey, Canada). Uncertainties on the hourly elemental concentrations result from the sum of independent uncertainties on: certified standards thickness (5%), aerosol collection area (2%), airflow (2%) and X-rays counting statistics (2–20%). Indeed, concentration values near to the minimum detection limits (MDL) have higher uncertainties. Typical detection limits range from about 10 ng m$^{-3}$ for low-Z elements down to 1 ng m$^{-3}$ (or below) for medium-high Z elements. The following 27 elements for a total sampling time of 500 h were detected in both fractions: Na, Mg, Al, Si, P, S, Cl, K, Ca, Ti, V, Cr, Mn, Fe, Ni, Cu, Zn, As, Se, Br, Rb, Sr, Y, Zr, Mo, Ba and Pb.

## 2.3 Positive Matrix Factorization

Positive Matrix Factorization (PMF) was applied to the hourly data sets (independently for fine and coarse fraction) allowing for identification of the major PM sources in both modes. In this study, the EPA PMF5.0 software was used. PMF is a widely used in air quality studies (e.g., Belis et al., 2019; Hopke et al., 2020) multivariate factor analysis model based on a weighted least square fit approach (Paatero and Tapper, 1994). It uses the uncertainties of each measurement to weigh the individual data points and imposes non-negativity constraints in the optimization process. In the PMF modelling procedure, a matrix of measurement data (X) is decomposed into two matrices to be determined: factor contributions (G) and factor profiles (F) as indicated in Eq. (1) (Paatero and Tapper, 1994):





$$X = G \cdot F + E ,\qquad(1)$$

where $X$, $G$ and $F$ are $n \times m$ matrix of the measurement data of $m$ chemical species in $n$ samples, $n \times p$ matrix of $p$ sources' contribution, and $p \times m$ matrix of profiles of $p$ sources, respectively, while $E$ is the residual matrix.

The mass balance equation (Eq. (1)) is solved in PMF by minimizing the object function Q given by Eq. (2) (Paatero, 1997):

$$Q = \left\| \frac{(X - G \cdot F)}{\sigma} \right\|_F^2 ,\qquad(2)$$

where $\sigma$ is the matrix of known uncertainties for measurement data.

According to the procedure of preparation of the input data proposed by Polissar et al. (1998), concentrations of elements below the detection limit (DL) were replaced with the values of 1/2 of the DL and their uncertainties were set at 5/6 of the DL values, while missing data were substituted by the geometric mean of the concentrations with uncertainties set at 4 times of the geometric mean concentration. The Signal-to-Noise (S/N) criterion (Paatero and Hopke, 2003) was applied to input data in order to separate the variables that retained a significant signal from that dominated by noise. Species with S/N > 2 were defined as "strong variables" and used in PMF as they are, species with 0.2 < S/N < 2 were classified as "weak variables" and downweighted by a factor of 3, while species with S/N < 0.2 were considered as "bad variables" and removed from the analysis. A criterion of the share of data above the DL was also used (Amato et al., 2016; Cesari et al., 2018). Finally, PMF was applied to 20 elements in 500 samples and 22 elements in 492 samples in the case of fine and coarse fraction, respectively. Based on the S/N ratio, most of the elements were considered as "good variables", except P, Cr, As, Se and Sr in the fine fraction, as well as P, As, Br, Zr, Ba and Pb in the coarse fraction, which were classified as "weak variables".

A number of solutions with 3 to 10 resolved factors were tested to find out the most optimal one. In order to examine the quality of the obtained solution several criteria were applied, including extracting realistic source profiles, distribution of scaled residuals and the comparison between the modelled and observed mass of elements. The best solution was obtained using 7 factors and 5 factors in the fine and coarse fraction, respectively, representing a reasonable physical interpretation of the sources with measured total concentration of elements reproduced well (with $R^2 = 0.96$ for fine fraction and $R^2 = 0.99$ for coarse fraction). Almost all variables showed scaled residuals estimated by PMF between -3 and +3. The bootstrapping method applied with 100 runs and minimum correlation R-value of 0.6 revealed stable PMF solution.

In this study, concentrations of PM mass and its macro components, i.e., organic carbon, elemental carbon, secondary inorganic aerosols and other water-soluble inorganic ions, were not available with the hourly temporal resolution. Therefore, the obtained results can be used for a detailed source identification but source time series will be expressed in arbitrary units (see e.g., Lucarelli et al., 2020).





### 2.4 Conditional Probability Function analysis

Conditional Probability Function (CPF) is a technique commonly used to identify the contributions of local and regional sources affecting a given monitoring site and the relationship between air pollutant concentrations and wind speed. The bivariate polar plots are used to illustrate the results of CPF, which calculates the probability that the concentration of a given species is greater than a specified value, usually expressed as a high percentile of the concentrations, as a function of both wind speed and direction, according to Eq. (3) (Uria-Tellaetxe and Carslaw, 2014):

$$CPF_{\Delta\theta} = \frac{m_{\Delta\theta}|_{C \geq x}}{n_{\Delta\theta}}, \tag{3}$$

where $m_{\Delta\theta}$ is the number of samples in the wind sector $\theta$ and having concentration C greater than or equal to a threshold value $x$, and $n_{\Delta\theta}$ is the total number of samples from wind sector $\Delta\theta$. Thus, CPF indicates the potential for a source region to contribute to high air pollution concentrations (Uria-Tellaetxe and Carslaw, 2014).

In this study, all CPF analyses were performed in the openair R package (Carslaw and Ropkins, 2012) and were applied to the PMF source factors assuming 90[th] percentile of the concentrations as a threshold value.

### 2.5 Air mass back trajectory clustering

The HYbrid Single-Particle Lagrangian Integrated Trajectory (HYSPLIT) model of the NOAA Air Resources Laboratory (Draxler and Hess, 1998) was applied to compute 96 h air mass backward trajectories starting at 200 m a.s.l. over the sampling site for each hour of the analyzed period. Meteorological fields from the ERA-Interim reanalysis (Dee et al., 2011) of the European Centre for Medium-Range Weather Forecasts (ECMWF) were formatted to be used as input data for the HYSPLIT runs. 6-hourly data are gridded in 27 pressure levels from 1000 hPa up to 100 hPa and were bilinearly interpolated to 0.5 deg horizontal resolution to take advantage of the 0.5 deg model's terrain.

Trajectories were classified into homogeneous groups by a non-hierarchical clustering procedure based on the k-means algorithm, which groups a given dataset into a number of clusters $k$ fixed a priori, assigning each case to the best fitting cluster. Each cluster is represented by its centroid, i.e., the average over the trajectories belonging to that cluster. Initially, $k$ starting centroids are set randomly with all trajectories allocated into the clusters of their nearest centroid. The centroids are then recalculated by averaging all the trajectories belonging to the same cluster in an iterative process until the cluster assignments no longer change. As the final clusters produced by the k-means algorithm are sensitive to the selection of initial centroids, for a given $k$ the clustering procedure was repeated 800 times in order to identify stable centroid positions and provide more robust results (Orza et al., 2012). The haversine formula of the great-circle distance between two points (Sinnott, 1984) given by Eq. (4) was used as the similarity measure in the clustering process:

$$D = 2R \sin^{-1}\left(\left[\sin^2\left(\frac{\phi_1 - \phi_2}{2}\right) + \cos\phi_1 \cos\phi_2 \sin^2\left(\frac{\lambda_1 - \lambda_2}{2}\right)\right]^{1/2}\right), \tag{4}$$





where $D$ is the distance between two points of the earth (km), $\varphi1$ and $\varphi2$ are their latitudes (radians), $\lambda1$ and $\lambda2$ are their

longitudes (radians), and $R$ is the Earth radius (6 367.45 km). The total distance between a trajectory $i$ and $a$ cluster centroid

$c$ is then:

$$D_{i,c} = \sum_{s=1}^{N_s} D_{i,c,s} \, , \qquad (5)$$

where $D_{i,c,s}$ is the distance between the points of back trajectory $i$ and centroid $c$ at the time step $s$, calculated with Equation

(4), and the summation runs over the total number of backward time steps. The score function that measures the quality of

the clustering is the total within-cluster Root Mean Squared Distance (RMSD) between individual trajectories and their

centroids:

$$RMSD_c = \sqrt{\frac{1}{N_c} \sum_{i=1}^{N_c} D_{i,c}^2} \, , \qquad (6)$$

$$RMSD_{total} = \sum_{c=1}^{k} RMSD_c \, , \qquad (7)$$

with $N_c$ the number of trajectories belonging to cluster $c$ and $k$ is the number of clusters. The optimal number of clusters was

assessed following a procedure similar to that of Dorling et al. (1992). The number of clusters $k$ was successively reduced by

one, from 30 down to 3 clusters, and the total within-cluster RMSD between individual trajectories and their centroids given

by Eqs. (6) and (7) was examined as a function of the number of clusters. The optimal number of clusters is, finally, the

lowest number of clusters for which the lowest percentage change in RMSD$_{total}$ is found when decreasing $k$ by one.

## 3 Results and discussion

### 3.1 Daily PM$_{2.5}$ composition and meteorological conditions

Simultaneously with the streaker measurement campaign, daily concentrations of PM$_{2.5}$ mass and its chemical composition,

including EC (elemental carbon), OC (organic carbon), water-soluble inorganic ions: $NO_3^-$, $SO_4^{2-}$, $NH_4^+$, $Cl^-$, $Na^+$, $K^+$, $Ca^{2+}$

and $Mg^{2+}$ were measured. PM$_{2.5}$ concentrations were determined according to the EN 12341:2014-07 standard: *Ambient air –*

*Standard gravimetric measurement method to determine the concentration of mass fractions PM$_{10}$ or PM$_{2.5}$ particulate*

*matter*. EC and OC content was determined with the use of Sunset Laboratory Thermal-Optical Carbon Aerosol analyzer,

equipped with flame ionization detector, using "EUSAAR_2" protocol (Cavalli et al., 2010), while the ionic constituents

were determined with the use of ion chromatography (Dionex ICS 1100, Thermo Scientific, USA) (see Juda-Rezler et al.,

2020).

The time series of daily concentrations of the main components of PM$_{2.5}$ throughout the measurement period (18$^{th}$ February

– 10$^{th}$ March 2016) is presented in Fig. 2.





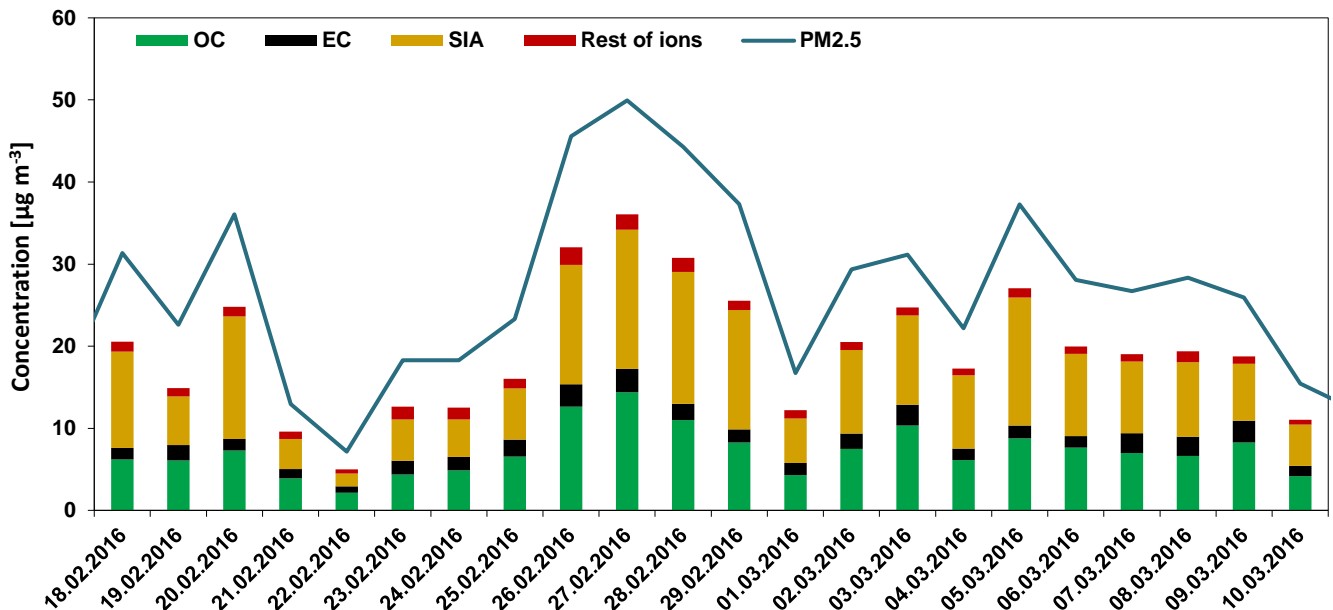

**Figure 2: Time series of daily concentrations (µg m⁻³) of PM₂.₅ (line) and its main components (bars) in Warsaw in the sampling period from 18th February to 10th March, 2016. Rest of ions indicate the sum of concentrations of K⁺, Cl⁻, Na⁺, Mg²⁺ and Ca²⁺ ions.**

Mean $PM_{2.5}$ mass concentration in the measurement period equaled 27.7 µg m⁻³ (SD = 11.0 µg m⁻³), being almost 50% higher that the annual mean $PM_{2.5}$ concentration observed in the whole 2016 (18.8 µg m⁻³). The highest value of 49.9 µg m⁻³ was recorded on February 27, exceeding almost 2 times the WHO daily air quality guideline (25 µg m⁻³). The main components of fine particulate matter in Warsaw were secondary inorganic aerosols (SIA, the sum of $SO_4^{2-}$, $NO_3^-$, $NH_4^+$) and organic carbon accounting for 35% and 26% of $PM_{2.5}$ mass, respectively. SIA and OC exhibited higher and lower

content, respectively, comparing to their contributions in $PM_{2.5}$ during the whole 2016, which in both cases reached on average about 30% of fine fraction (see Juda-Rezler et al., 2020). At the same time, the contribution of these components exceeds their typical content in $PM_{2.5}$ mass reported for urban background sites in Southern and Central Europe (see e.g., Amato et al., 2016; Błaszczak et al., 2019). Among SIA, the highest share was observed for $SO_4^{2-}$ (17% and 49% in $PM_{2.5}$ and SIA mass, respectively), followed by $NO_3^-$ (12% and 34%) and $NH_4^+$ (6% and 17%). The share of the remaining $PM_{2.5}$

components was much smaller: EC constitutes about 6% and inorganic ions other than SIA (K⁺, Cl⁻, Na⁺, Mg²⁺, Ca²⁺) – around 4% of $PM_{2.5}$. As reported by Juda-Rezler et al. (2020), in the heating season during which the "streaker" measurement campaign has been carried out, a remarkable increase of concentrations of OC, EC, $NO_3^-$, $NH_4^+$ and Cl⁻ was observed in comparison to other seasons.

During the measurement campaign the mean values of temperature, relative humidity, air pressure, wind speed, predominant

wind direction and sum of precipitation measured at sampling site were 3 °C, 87%, 1009 hPa, 2 m s⁻¹, W and E, and 24 mm, respectively.





## 3.2 Hourly elemental composition

The descriptive statistics for the hourly concentrations of 27 analyzed elements measured in both size fractions are given in Table 1.

**Table 1: Descriptive statistics for the hourly concentrations (ng m$^{-3}$) of the elements measured in fine and coarse fractions.**

| Element [ng m$^{-3}$] | Fine | | | | | Coarse | | | | |
|---|---|---|---|---|---|---|---|---|---|---|
| | Mean | Median | Min | Max | SD | Mean | Median | Min | Max | SD |
| Al | 25.3 | 18.9 | 3.9 | 206.0 | 24.1 | 70.2 | 50.9 | 2.2 | 477.6 | 64.9 |
| As | 0.5 | 0.3 | 0.1 | 2.3 | 0.4 | 0.2 | 0.1 | 0.04 | 0.9 | 0.1 |
| Ba | 7.8 | 6.5 | 5.1 | 28.1 | 3.7 | 6.5 | 3.4 | 2.1 | 42.9 | 5.4 |
| Br | 2.4 | 2.2 | 0.4 | 9.7 | 1.3 | 0.2 | 0.1 | 0.05 | 3.0 | 0.2 |
| Ca | 48.8 | 31.5 | 4.1 | 287.2 | 45.4 | 108.1 | 84.7 | 4.3 | 449.8 | 88.8 |
| Cl | 113.3 | 60.1 | 5.5 | 1 492.9 | 168.1 | 141.8 | 35.5 | 1.5 | 1 682.5 | 251.3 |
| Cr | 2.5 | 2.3 | 0.1 | 14.4 | 1.3 | 1.3 | 1.1 | 0.2 | 6.9 | 0.9 |
| Cu | 6.7 | 4.5 | 0.7 | 190.2 | 13.2 | 4.9 | 3.6 | 0.1 | 41.3 | 4.8 |
| Fe | 114.7 | 95.4 | 13.2 | 1 296.9 | 105.1 | 193.1 | 154.3 | 11.1 | 1 281.7 | 148.8 |
| K | 166.8 | 153.2 | 18.1 | 639.6 | 85.1 | 29.9 | 24.5 | 2.1 | 152.9 | 20.1 |
| Mg | 17.4 | 17.0 | 4.9 | 48.4 | 8.6 | 36.8 | 25.4 | 3.0 | 207.3 | 33.5 |
| Mn | 2.4 | 2.1 | 0.4 | 21.1 | 1.9 | 2.3 | 2.0 | 0.2 | 11.8 | 1.6 |
| Mo | 0.4 | 0.4 | 0.2 | 2.1 | 0.2 | 0.2 | 0.2 | 0.1 | 0.9 | 0.1 |
| Na | 80.4 | 57.2 | 8.6 | 1 538.5 | 91.8 | 170.6 | 71.7 | 3.5 | 1 347.8 | 231.5 |
| Ni | 0.8 | 0.7 | 0.1 | 8.2 | 0.6 | 0.3 | 0.2 | 0.02 | 2.4 | 0.2 |
| P | 20.9 | 19.8 | 1.8 | 48.7 | 8.3 | 7.0 | 6.5 | 1.3 | 21.7 | 3.3 |
| Pb | 11.7 | 9.9 | 0.5 | 146.2 | 11.5 | 1.0 | 0.3 | 0.2 | 20.5 | 1.5 |
| Rb | 0.3 | 0.2 | 0.1 | 3.1 | 0.2 | 0.2 | 0.1 | 0.1 | 10.7 | 0.5 |
| S | 1 020.8 | 907.6 | 185.2 | 2 612.8 | 548.0 | 73.9 | 55.7 | 6.1 | 468.0 | 62.2 |
| Se | 0.4 | 0.4 | 0.1 | 1.2 | 0.2 | 0.1 | 0.1 | 0.05 | 0.5 | 0.04 |
| Si | 51.4 | 51.4 | 0.7 | 349.1 | 39.2 | 166.1 | 126.4 | 8.9 | 995.5 | 134.4 |
| Sr | 0.5 | 0.3 | 0.2 | 15.8 | 0.9 | 0.4 | 0.3 | 0.1 | 3.1 | 0.4 |
| Ti | 2.9 | 1.7 | 1.3 | 20.6 | 2.4 | 5.7 | 4.4 | 0.6 | 34.6 | 4.8 |
| V | 1.2 | 1.0 | 0.8 | 3.3 | 0.5 | 0.6 | 0.4 | 0.3 | 2.5 | 0.3 |
| Y | 0.3 | 0.3 | 0.2 | 2.3 | 0.1 | 0.1 | 0.1 | 0.1 | 1.1 | 0.1 |
| Zn | 38.7 | 33.0 | 6.7 | 239.5 | 25.3 | 6.5 | 4.5 | 0.4 | 52.2 | 6.0 |
| Zr | 0.4 | 0.3 | 0.2 | 2.2 | 0.2 | 0.4 | 0.2 | 0.1 | 3.3 | 0.4 |

In terms of PM$_{10}$ (the sum of fine and coarse fraction), S (1 095 ng m$^{-3}$) is found to be the most abundant element, followed by Fe (308 ng m$^{-3}$), Cl (255 ng m$^{-3}$), Na (251 ng m$^{-3}$), Si (218 ng m$^{-3}$), K (197 ng m$^{-3}$) and Ca (157 ng m$^{-3}$). The concentrations lower than 1 ng m$^{-3}$, and in some cases below the detection limits, are observed for As, Mo, Rb, Se, Sr, Y and 260 Zr. The analyzed elements may be divided into three groups according to their presence in the respective fraction. The first





group represents the elements which are abundant mainly in the fine fraction, with both mean and median values substantially higher than in the coarse one. These are As, Br, Cr, Cu, K, Mo, Ni, P, Pb, S, Se, V, Y and Zn, all of them being typically emitted by fuel combustion and road traffic (e.g., Belis et al., 2013). Metals being primarily of crustal or marine origin, i.e.: Al, Ca, Fe, Mg, Na, Si and Ti belong to the second group of elements distributed mainly in the coarse fraction.

Different behavior is observed for Cl which content in coarse fraction is characterized by higher mean value but lower median value comparing to the fine fraction. In the third group, the concentrations of Ba, Mn, Rb, Sr and Zr, being the elements related with the abrasion processes as tire and brake wear (Pant and Harrison, 2013), are balanced between the two fractions.

The hourly time resolution of elements measured in the fine and coarse fractions allowed to follow in detail diurnal evolution

of primary PM sources activity and formation of secondary aerosol. The analysis of concentration time series revealed the differences between the two fractions for most of the elements. As an example, the temporal trends of selected elements contained in both fractions are shown in Fig. 3, while the trends for the rest of elements are given in Supplementary material (Figs. S1-S23). The following paragraphs present discussion of the time series of four selected elements, whose concentrations display various patterns and help distinguishing and identifying PM emission sources.






**Figure 3: Hourly concentrations (ng m$^{-3}$) of Cl, Cr, K and Al measured in the fine (green) and coarse (red) fractions.**





Cl is usually attributed either to the sea salt in areas close to the coasts or to the road salt in continental areas of Central and Northern Europe (Belis et al., 2013). The latter case is the most probable in Warsaw. Recorded time series of Cl in Warsaw are however different in the fine and coarse fractions with no correlation between the concentrations in the two modes ($r$ = 0.08). In the coarse fraction, Cl is mainly correlated with other markers of road salt, i.e.: Na ($r$ = 0.96) and Mg ($r$ = 0.78),

while in the fine fraction Cl is not correlated with other elements ($r$ = 0.02 – 0.44), except Br and K for which moderate correlations ($r$ = 0.69 and $r$ = 0.54, respectively) are observed. Concentrations of Cl are higher in the coarse fraction, with mean level equals to 142 ng m$^{-3}$ and several peak values up to almost 1 700 ng m$^{-3}$, while in the fine one mean Cl level reaches 113 ng m$^{-3}$, with peaks up to almost 1 500 ng m$^{-3}$ (Fig. 3). In the coarse fraction a strong increase of Cl is always present together with Na and Mg peaks, while in the fine fraction during every increase of Cl the increase of concentrations

of diverse elements is observed and most often it is accompanied by slight increase of the concentrations of Br, K or different crustal elements. The temporal trend of Cl in the coarse fraction is characterized by peaks in the morning (between 06:00 and 08:00) and evening (between 17:00 and 23:00). On the contrary, there is no clear daily pattern of Cl in the fine fraction, where the peaks of elevated concentrations last a few hours and occur on different days of the week and at different time of the day (Fig. 3). This suggests different origin of Cl in both fractions.

There is also no correlation between the concentrations in the two fractions ($r$ = 0.04) in the case of Cr, being generally associated with industrial or vehicular emissions (e.g., Taiwo et al., 2014b; Banerjee et al., 2015), but also with coal burning in small domestic boilers (Juda-Rezler et al., 2011). In the fine fraction, Cr is moderately correlated with Ni ($r$ = 0.56). For other elements a weak or no correlation is observed, yet the highest $r$ values were obtained for other markers of industrial activities, i.e.: Fe ($r$ = 0.49), Mn ($r$ = 0.42) and Mo ($r$ = 0.41). In the coarse fraction, Cr is strongly correlated with Fe ($r$ =

0.82), Mn ($r$ = 0.77) and Cu ($r$ = 0.75) and moderately with other traffic related elements, i.e., Ba ($r$ = 0.68), Zn ($r$ = 0.50), Ni ($r$ = 0.50) as well as with crustal elements, such as Si ($r$ = 0.59), Ca ($r$ = 0.57) and Ti ($r$ = 0.52). This may suggest an industrial and traffic origin of Cr in the fine and coarse fraction, respectively. Concentrations of Cr are almost 2 times higher in the fine fraction (mean value: 2.3 ng m$^{-3}$) than in the coarse one (1.3 ng m$^{-3}$). As can be seen in Fig. 3, concentrations of Cr in the fine fraction show no diurnal variation and display several peaks up to 14.4 ng m$^{-3}$, while in the coarse fraction a

bimodal diurnal cycle with typical peaks in the traffic rush hours from 06:00 to 10:00 and from 15:00 to 21:00 is observed. Conversely to both previous elements, a weak correlation between the concentrations in the fine and coarse fraction ($r$ = 0.40) is observed for K, which is present in the mineral dust but is also a typical marker of biomass burning (e.g., Nava et al., 2015). Concentrations of K are substantially higher in the fine fraction (mean value: 167 ng m$^{-3}$) than in the coarse one (30 ng m$^{-3}$). In the fine fraction, K is moderately correlated with elements recognized as markers of coal and wood combustion

(e.g., Belis et al., 2013; Nava et al., 2015), i.e.: Br ($r$ = 0.62), Se ($r$ = 0.60), Zn ($r$ = 0.58), S ($r$ = 0.55) and Cl ($r$ = 0.54). In addition, the diurnal cycle of concentrations of K in this fraction is characterized by peaks in the morning and evening hours (Fig. 3), which clearly point on solid fuels combustion in residential sector. In the coarse fraction, K is mainly correlated with typical crustal elements, showing a strong correlation with Al ($r$ = 0.81), Si ($r$ = 0.80), Ti ($r$ = 0.76) and moderate with Ca ($r$ = 0.69), Sr ($r$ = 0.63), Fe ($r$ = 0.62) and Mg ($r$ = 0.60). There is no clear daily variation of K in this fraction, however





daytime concentrations slightly exceed nighttime levels, which is likely related to the resuspension of mineral dust during daytime activities. Several peaks of concentrations of K, which are always present together with other crustal elements, are present during the measurement period.

A completely different behavior can be observed for Al, which together with Ca, Fe, Mg, Si, Sr and Ti, is a typical marker of mineral dust (e.g., Banerjee et al., 2015). Concentrations of Al are almost 2.5-times higher in the coarse fraction (mean

value: 70 ng m$^{-3}$) than in the fine one (25 ng m$^{-3}$). A strong correlation between the concentrations in the fine and coarse fraction ($r = 0.70$) is observed for this element. Moreover, in both modes Al is correlated only with crustal elements, i.e., Ti ($r = 0.81$), Si ($r = 0.63$) and Ca ($r = 0.51$) in the fine fraction, as well as Si ($r = 0.92$), Ti ($r = 0.89$), K ($r = 0.81$), Ca ($r = 0.65$), Sr ($r = 0.62$) and Fe ($r = 0.55$) in the coarse one. There is also similar diurnal variation in both fractions, with Al concentrations higher during the daytime than in the nighttime. Most of the peak values of Al are present in both modes in

the same time (Fig. 3) and occur together with other crustal elements, which suggest crustal origin of the element in both fractions.

**3.3 PM source apportionment**

As it was described in Section 2.3, a solution with 7 and 5 factors has been chosen for the fine and coarse fraction, respectively. The identified source profiles together with daily patterns of the sources are shown in Figs. 4 and 5. The

following similar sources were observed in both fractions: sulfates, soil dust, road salt, traffic- and industry-related sources, while Cl-rich and wood and waste combustion sources were identified only in the fine fraction. A detailed description of the identified sources is presented in the following sections 3.3.1 – 3.3.7.

For better interpretation of the identified sources, the polar plots of CPF analysis, combining PMF results with wind speed and direction, are presented in Fig. 6.







**Figure 4: Left panel: PMF profiles (bars, left y axis) and explained variations (black diamonds, right y axis) of the identified sources for the fine fraction. Right panel: Daily patterns of the identified sources (in arbitrary units).**




**Figure 5: Left panel: PMF profiles (bars, left y axis) and explained variations (black diamonds, right y axis) of the identified**
**sources for the coarse fraction. Right panel: Daily patterns of the identified sources (in arbitrary units).**






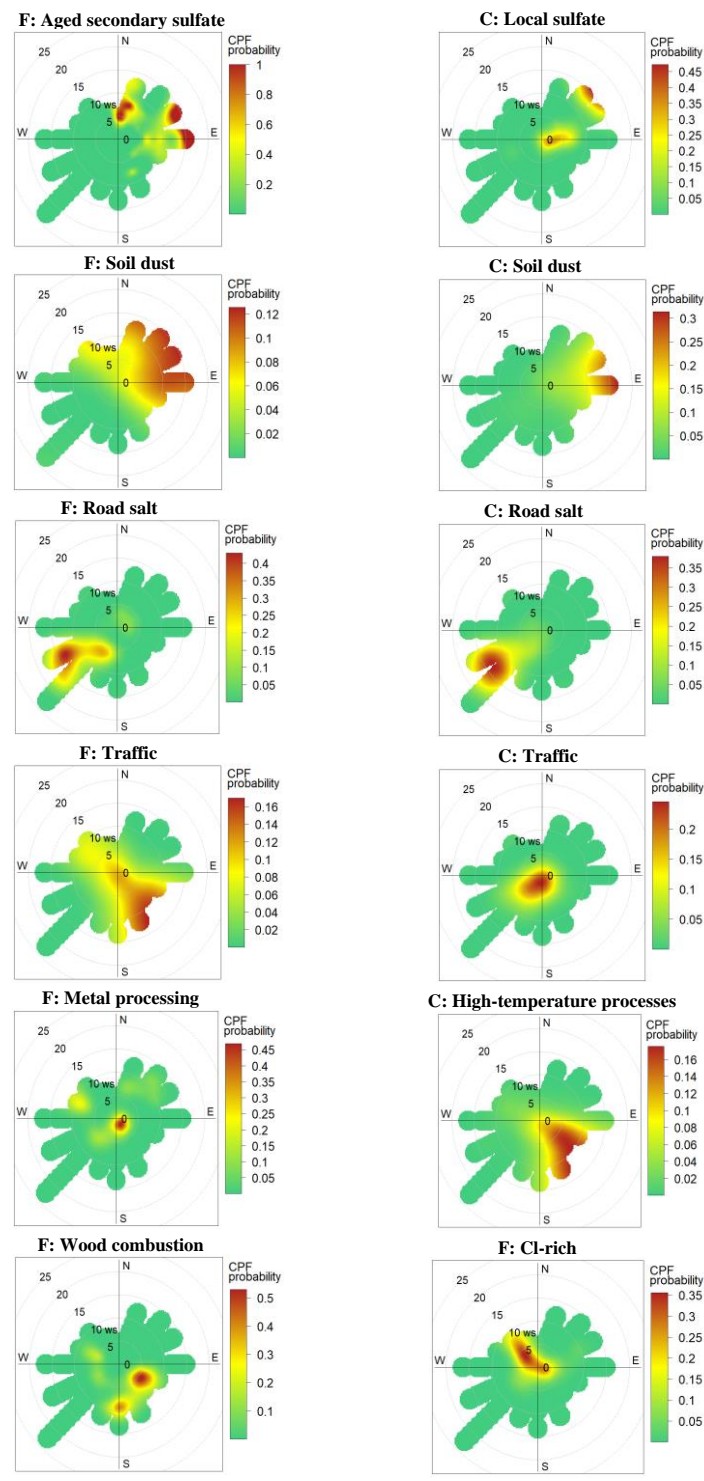

**Figure 6: Conditional Probability Function (CPF) analysis (at 90th percentile) of the PMF sources identified in the fine (F) and coarse (C) fractions. Wind speed (ws) is expressed in km h$^{-1}$.**



### 3.3.1 Sulfates

In both fractions the factor is characterized by high contributions of sulfur, representing 67% and 83% of the total mass of S in the fine and coarse fraction, respectively (Figs. 4 and 5). In addition, in the fine fraction the factor is associated with
notable loadings of P (30% of P mass), Ni (23%) and K (22%) and, to a lesser extent of Se and Br (around 15% of the elements mass). In the coarse fraction the factor exhibits similar profile with notable loadings of P (31%), Br (23%), K (18%) and Ni (17%). Se in this fraction was excluded from the PMF analysis as having the S/N ratio < 0.2 (see Section 2.3), however 12% of the total mass of another marker of coal combustion – As – appears in the source profile. It is noteworthy that in wintertime in Warsaw, As and K appeared to be highly bioavailable to the living systems, with increasing
bioavailability during the days of elevated air pollution levels, suggesting a higher risk to humans posed by emission from this source during those days (Juda-Rezler et al., 2021).

The presence of sulfur is associated with a secondary component produced by combustion processes of sulfur-containing fossil fuels. In the case of this study its most probable source is domestic heating, emitting substantial amount of $SO_2$, which in turn forms sulfate in the aqueous-phase conversion (Seinfeld and Pandis, 2016). All other elements related with this
factor, as well as their mixture, have been previously attributed to small-scale residential combustion. K is a well-known marker of biomass burning, while Br may be produced by this type of sources as well (Nava et al., 2015). Also high contribution of P (together with K) has been previously attributed to the wood combustion for heating purposes (Richard et al., 2011). Although As and Se are recognized as markers of coal combustion in power plants (e.g., Banerjee et al., 2015), a substantial amount of these elements can also be released when coal is burnt in residential sector (IARC, 2012; Bano et al.,
2018; Zhao and Luo, 2018). In turn, oil combustion is the main source of Ni (e.g., Belis et al., 2013).

The S/K ratio can be used as an indicator to describe the rate of accumulation of S compounds in biomass burning aerosols during their transport. The ratios may range from 0.5 for fresh emissions up to 8 for transported and aged aerosol (Viana et al., 2013 and references therein). The S/K ratios obtained in this study equal 6.1 and 2.5 for the fine and coarse fraction, respectively, pointing on aged fine aerosols and more freshly emitted coarse ones.
In general, secondary sulfate is primarily formed in the accumulation mode (particles with aerodynamic diameter 0.1–1 μm). Since secondary aerosol formation proceeds relatively slowly (in the time frame of hours and days), the daily source profile in the fine fraction does not exhibit a clear diurnal variation (Fig. 4). In the case of coarse fraction, sulfate formation under nighttime condition of high relative humidity can play an important role. During the measurement period the relative humidity was high, with average values of 89.4% and 84.2% at nighttime and daytime, respectively. Under such conditions,
$SO_2$ oxidation to sulfate occurs mainly on the particles with water film surface or in the aqueous phase of particles. The oxidation is catalyzed by transition metals, in particular Fe and Mn (see e.g., Sarangi et al., 2018; Wang et al., 2020). Therefore the pattern of source profile in this fraction is different than in the fine one, showing higher concentrations during late afternoon (around 15:00) and night (starting from 17:00), as well as nighttime contributions 2 times higher than those registered during daytime (Fig. 5).


This confirms that the fine fraction is dominated by regional rather than local transport of $SO_2$ and sulfate from Warsaw's outskirts with individual residential heating, mainly with the use of biomass and solid fuels, whereas in the coarse fraction the presence of sulfate is probably due to the local combustion activities in the city itself. In addition, CPF analysis (Fig. 6) indicates the highest contributions at high wind speeds from regional sources to the North and East, where the individual heating is located, but also from local sources in the case of coarse fraction. Thus, the source was recognized as "Aged

secondary sulfate" and "Local sulfate" for the fine and coarse fraction, respectively.

Overall, this source accounts for the highest contribution to the fine elemental mass (mean share during the measurement period equals 44%) and notable contribution to the coarse elemental mass (11.5%). This result suggests that regional sources including emissions from the residential sector located to the North and East of Warsaw, have a significant influence on the fine PM composition in the city. This is consistent with the conclusions of Juda-Rezler et al. (2020) who, based on the

analysis of daily concentrations of $PM_{2.5}$ and its constituents measured in Warsaw for the whole 2016 year, found that fresh and aged aerosol from the residential sector transported from the outskirts of the city constitutes on average 45% of $PM_{2.5}$ mass.

### 3.3.2 Soil dust

This factor, accounting for the highest contribution to the coarse elemental mass (31.5%) and notable contribution to the fine

elemental mass (13.5%), is characterized by contributions from the typically crustal elements (e.g., Belis et al., 2013; Banerjee et al., 2015), i.e.: Si (91% of Si mass), Ca (71%), Al (41%), Fe (21%) and Mg (19%) in the fine fraction, as well as Al (77%), Si (71%), Ti (63%), K (52%), Ca (47%), Sr (31%) and Mg (29%) in the coarse one. In addition, in the coarse fraction along with crustal elements, notable contributions of species associated with traffic emissions (see Section 3.3.4), i.e.: Ni (30%), Mn (23%), P (21%), Ba (19%), Zr (15%) and Cr (14%) are noted. Therefore, in this fraction the source can be

assigned to mixed source, consisting of soil dust with substantial contribution of road dust.

Diurnal profiles in both size fractions have comparable patterns (Figs. 4 and 5) with substantially higher concentrations during the day (between 06:00 and 19:00), however the peak values observed in the coarse fraction overlap with morning and afternoon traffic rush hours (07:00-8:00 and 16:00), confirming road dust as a contributing source. CPF analysis (Fig. 6) shows that, despite some differences in the source profile and diurnal variation in both cases, the highest contributions are

from the Northeast at high and moderate wind speeds and from the East at high wind speeds, suggesting similar source locations.

### 3.3.3 Road salt

This is the second largest factor contributing to the coarse elemental mass (31%) and noticeable one in the case of the fine elemental mass (12.5%), and is characterized by high loading of elements attributed to the road salt. During the measurement

campaign deicing salt, mainly composed of NaCl and/or MgCl, has been applied on the roads in Warsaw. While Na and Mg have notable contributions in both fractions (Na: 71% and 77%; Mg: 44% and 29% in the coarse and fine fraction,





respectively), the two sources can be differentiated based on the level of Cl (Figs. 4 and 5). This factor reconstructs >80% of Cl mass in the coarse fraction and only 5% in the fine one, indicating fresh coarse and aged fine particles with a road salt origin. The Cl depletion in the fine fraction is most probably because of the well-known heterogeneous chemical reactions
between NaCl and nitric ($HNO_3$) and sulfuric ($H_2SO_4$) acids resulting in formation of sodium nitrate and sodium sulfate as well as in volatilization of HCl (Seinfeld and Pandis, 2016). In addition, the factor in the fine fraction includes also notable contributions from elements usually associated with traffic emissions (P, Cr, Ni, Br), which confirms the anthropogenic character of the source.

As can be seen in Fig. 5, in the coarse fraction daytime levels exceed the nighttime ones with peak values observed in the
morning and afternoon traffic rush hours (08:00-11:00 and 16:00), indicating that the road salt is resuspended through daytime activities nearby the measurement site. In the fine fraction, the diurnal variation is less evident and less variable (Fig. 4), however some peaks are observed in similar traffic rush hours (06:00-11:00 and slight increase starting from 16:00), which may suggest both local origin of the salt and its transport from distant roads nearby the city. CPF analysis (Fig. 6) indicates that the highest contributions of both source profiles are from the Southwest at high and moderate wind speeds, and
in addition at lower wind speeds in the case of coarse fraction, suggesting similar source locations. In this direction one of the main city's roads as well as an expressway are located.

### 3.3.4 Traffic emissions

In the fine fraction, this factor, explaining 8% of the elemental mass, is associated with high loadings of Fe (66% of Fe mass), Mn (42%), Ni (28%), Cr (21%), and small amounts of Zn (14%), Si (12%) and Pb (9%). In the coarse fraction, the
factor contributes to the elemental mass in 20% and is dominated by Cu and Fe, with 76% and 63% of the mass explained by these elements, respectively. High loadings of Cr (42%), Mn (36%), Ca (30%), Zr (30%), Ba (28%), and to a lesser extent of Br (16%) and Ni (13%) are present as well. Cu in the fine fraction appears not to be associated with traffic emissions (see Section 3.3.5). All the above mentioned elements may be emitted from abrasion of roads (Ca, Fe, Mn, Si), brake linings and pads (Ba, Cu, Cr, Fe, Mn, Pb, Zr), as well as of tires (Fe, Mn, Zn) (e.g., Pant and Harrison, 2013; Amato et al., 2014;
Banerjee et al., 2015). Nevertheless, most of these elements can also be related with diverse exhaust-related emissions, i.e., fuel and lubricant combustion, catalytic converters, particulate filters and engine corrosion (Pant and Harrison, 2013 and references therein), while Ba can be emitted from gasoline, liquefied petroleum gas and diesel engines. Ba and Zn have been reported to be strongly associated with diesel fuel, while Cu, Mn and Sr with gasoline (Pant and Harrison, 2013). Fe, being a fuel additive, can be emitted from diesel engines (Bugarski et al., 2016). Thus, the factor in both fractions is likely to be
attributed to the more general source "Traffic", including both non-exhaust and exhaust emissions.

The factor in both modes displays a strongly bi-modal cycle that followed times of peak characteristic for traffic (Fig. 4 and 5). The highest levels are observed in the morning (between 06:00 and 10:00 with peaks at 07:00-8:00) and late afternoon and evening (between 15:00 and 21:00 with peaks at 18:00), although in the coarse fraction the second peak is less evident. CPF analysis (Fig. 6) shows that the highest source contributions for the two fractions come from different directions. For



the fine fraction, the highest contributions are from the Southeast at high and moderate wind speeds with lower probability
       for low wind speeds corresponding with the locations of another main city's road, while the highest contributions for the
       coarse fraction come only at low wind speeds originating from the nearby roads.

### 3.3.5 Industrial processes

       Overall, this factor does not represent a significant emission source influencing the air quality in Warsaw, as it accounts for
2% and 6% of the total fine and coarse elemental mass, respectively. In the fine fraction, the factor is dominated solely by
       Cu (66% of Cu mass) and Pb (16%), while over 90% of the mass of Zn and not negligible parts of mass of Mn (25%), P
       (17%), Pb (16%), Cr (14%), Ni (14%), As (10%) and Fe (10%) are found in the coarse fraction. These elements are
       commonly recognized as being emitted from industrial processes, such as ferrous and non-ferrous metal processing, and steel
       industries (e.g., Banerjee et al., 2015; Amato et al., 2016). In addition, As, Cu, Cr and Sn (which was not determined in the
present study) have been found as specific tracers of glassmaking emissions (Ledoux et al., 2017). As it was mentioned
       above, As is a well-known marker of coal combustion in power industry.
       No specific diurnal variation with episodic peaks at different time of the day can be seen for the fine mode (Fig. 4). The time
       series of the factor shows intense contribution of the elevated concentrations occurring in the night (between 01:00 – 04:00)
       and smaller peaks at 07:00 and 19:00. During the day almost stable pattern is exhibited. The nighttime peak is mainly a
result of elevated values recorded on 24th February, however the causes of this episode were not identified. The CPF
       analysis (Fig. 6) indicates two possible sources of the factor. The first one, having the highest contributions to peak levels, is
       located nearby the monitoring site and may be associated with welding activities carried out in the area of water treatment
       station where the measurement site was located. Cu and Pb have been found among the main components of welding
       emissions (e.g., Golbabaei and Khadem, 2015). The second source, with lower probability of high factor levels is related
with moderate wind speeds and is placed in the Northwest. One of the biggest Polish steelworks with an electric arc furnace
       is located in this direction, around 10 km from the sampling site. A steelworks can also be associated with substantial
       emissions of Cu and Pb (e.g., Yatkin and Bayram, 2008). Thus, the source can be classified as a mixed and named "Metal
       processing".
       The time series of the factor in the coarse fraction is characterized by a higher number of episodic peaks, also suggesting an
industrial source (Fig. 5). Based on the high contributions of Zn and other industry-related elements, the source may be
       classified as industrial emissions from steelworks and/or glassworks. Some amounts of As may also suggest the coal
       combustion in a power plant. The diurnal cycle demonstrates also clear peaks at 06:00, 10:00 and 18:00 which may point
       also the influence from a traffic source, however further separation of the sources was not possible. The CPF analysis (Fig.
       6) points the highest contributions from the Southeast at high and moderate wind speeds. In this direction a combined heat
and power plant is located within the city's borders. There is no typical industrial plants located southeasterly in Warsaw,
       however, at a distance of up to 100 km at which coarse particles can be typically transported (Seinfeld and Pandis, 2016)



some industrial facilities such as steelworks, glassworks and another coal-fired power plant are located. Therefore, in this fraction the source can be classified as "High-temperature processes".

### 3.3.6 Wood combustion

This factor, identified only in the fine fraction (Fig. 4), contributes to the elemental mass in 10% and is dominated by Zn (54% of Zn mass), Pb (48%) and K (40%) with smaller contributions of Se (23%), Br (22%) and As (10%). As was already mentioned, K is the major compound emitted from wood biomass burning, whereas Pb and Br can be emitted during waste wood combustion and biomass burning, respectively (Nava et al., 2015). In addition, the combustion of wood pellets can generate, besides K, also high amounts of Zn, Pb and As (Vicente et al., 2015). Therefore, this factor can be classified as "Wood combustion". However, since As and Se are well known markers of coal combustion in power industry, the biomass co-firing in large point sources cannot be excluded.

The temporal trend of this source follows the typical daily pattern of fuel combustion for the heating purposes with the maximum concentrations in the morning and evening, as well as during the nighttime when the heat demand is the highest (Fig. 4), supporting its interpretation as a wood combustion for residential heating. The CPF analysis (Fig. 6) points two possible sources of the factor. The first one shows the highest contributions to peak levels from South-east at moderate wind speeds. In this direction a combined heat and power plant co-firing hard coal and wood biomass is located. Moreover, in this plant a new biomass-fired boiler, with the wood biomass as basic fuel, was commissioned in 2016 and during the measurement period the boiler trial operation was carried out. The second source is located in the South and the highest contributions are associated with higher wind speeds. This direction covers the districts with less dense Warsaw's heat network and thus may suggest a wood combustion contribution to the factor.

### 3.3.7 Cl-rich

The second factor identified solely in the fine fraction is dominated by Cl, with 78% of the mass explained by this element (Fig. 4). Although Cl is associated with sea/road salt or with waste combustion, this factor does not correlate neither with salt-related nor combustion-related species. However, the source profile contains small amounts of Br (14% of Br mass), Pb (8%), K (7%), Se (6%), S (6%) and Zn (5%), which may point to waste combustion (e.g., Banerjee et al., 2015). On the other hand, the CPF analysis (Fig. 6) shows the highest contribution from the North-west direction at all range of wind speeds, where emission sources of this type are not located. Instead, there are some facilities where substantial amount of chlorine based chemicals are used to maintenance of the infrastructure, as well as many small car repair workshops where some amount of wastes could be combusted. In addition, the factor shows no clear diurnal variation with episodic peaks at different time of the day and lasting for a few hours. Thus, the precise identification of the emission source is not possible, although the factor contributes to the fine elemental mass in 10%.





### 3.3.8 Bootstrap

The chosen solution represents a reasonable physical interpretation of the sources with satisfactory reproduction of the measured total elemental concentration. Moreover, the bootstrap (BS) analysis (100 runs, 0.6 minimum correlation R-value)

has shown that for 3 out of 7 sources identified in the fine fraction ("Aged secondary sulfate", "Cl-rich", "Road salt") and for 3 out of 5 in the coarse fraction ("Soil dust", "Road salt", "Traffic") all BS factors were assigned to base case factors in 100% of every BS resample. For the rest of the sources the criteria of overall reproducibility suggested by EPA PMF guide, i.e., 80%, have been also met. In the case of "Soil dust" and "Metal processing" factors in the fine fraction, 3% and 2% of BS factors, respectively, were not assigned to any of base case factor. Furthermore, no factor swaps for any values of dQmax

have been found in the displacement analysis, implying that the PMF solution was well-defined.

### 3.4 Air mass back trajectories

The cluster analysis of back trajectories identified seven air flow types, whose representative trajectories (centroids) arriving at 200 m a.s.l. together with the contributions of the identified by PMF sources for fine and coarse fractions are shown in Fig. 7.

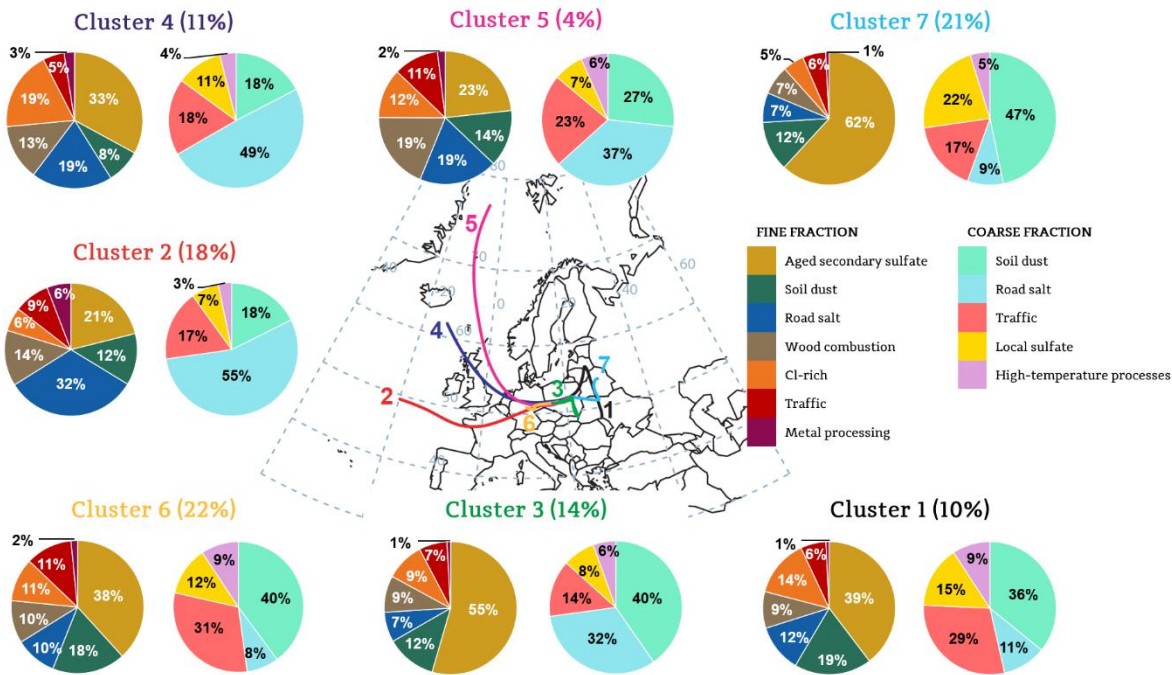


**Figure 7: Trajectory cluster centroids arriving to Warsaw at 200 m (center map) with PMF factor contribution in different clusters for fine (left pie charts) and coarse (right pie charts) fraction. Percentage of the trajectories classified into each cluster is given in the parentheses.**



The most frequent advection patterns correspond to the regional slow moving air masses, which together account for 67% of the total number of trajectories reaching the measurement site. Such flows, including short westerly trajectories (Cluster 6) and trajectories recirculating over Eastern European countries (Clusters 1 and 7) and over Poland (Cluster 3), are characterized by the shortest trajectories (the clusters length between 1 900 km and 2 100 km) and the net distance travelled by an air parcel within 96 h between 550 km and 850 km. Cluster 6, being the major cluster in terms of the number of

trajectories (22% of the trajectories), originates from central part of Germany and passes further over eastern Germany, as well as over western and central Poland. The second largest cluster in terms of the number of trajectories (21%) is Cluster 7, whose origin is in eastern Belarus. The air masses pass further over Belarus-Ukraine border and reach Warsaw from the East. 10% of the trajectories are identified in Cluster 1, which starts over southern Ukraine and arrive to Warsaw from the Northeast, passing over Ukraine, Belarus, Latvia and Lithuania. Cluster 3 (14% of the trajectories), is composed of the

trajectories recirculating over western and southern Poland.

One third of the trajectories correspond to fast north-westerly (Clusters 4 and 5) and westerly (Cluster 2) flows (Fig. 7), with the length of trajectories between 2 800 km and 4 025 km. Air masses in Clusters 2 and 4, accounting for 18% and 11% of the trajectories, respectively, originate from the Atlantic Ocean and pass over western European countries, wherein air masses in Cluster 4 are moving more from the Northwest direction. Cluster 5 being the smallest in terms of the number of

trajectories (4%) is composed of the trajectories starting from the Greenland Sea and passing over the Norwegian Sea and the North Sea. All three clusters after crossing central part of Germany arrive to Warsaw from the West, passing over western and central Poland.

The analysis of the influence of air mass origin on PMF factor contributions shows that slow and regional flows are related to higher concentrations of the fine fraction than the coarse one, while the fast moving westerly flows show comparable

levels for both fractions. In addition, regional air masses are characterized by the highest contributions of "Aged secondary sulfate" to the fine elemental mass, accounting for 38%, 39%, 55% and 62% for Clusters 6, 1, 3 and 7, respectively. This confirms the regional transport of aged aerosol to the sampling site, in particular from the East (Cluster 7) and South (Cluster 3). The share of the rest of identified sources is almost equally distributed within the clusters, pointing on more local origin of the sources. In the case of fast Atlantic air masses, the contribution of sources to the elemental mass is different. Clusters

4 and 5 are characterized by the highest contribution of "Aged secondary sulfate" (33% and 23%, respectively), followed by "Road salt" (19% in both clusters), while on the contrary Cluster 2 has the highest contribution of "Road salt", followed by "Aged secondary sulfate" (32% and 21%, respectively). This may suggest that although road salt is the main source of atmospheric aerosol in Warsaw, the partial contribution of sea spray cannot be neglected. For the coarse fraction, the slow regional air masses are characterized by substantially higher contributions of "Soil dust" (36 – 47%) than the fast westerly

ones, while the fast Atlantic air masses bring substantially higher loads of the "Road salt" (shares equal 37 – 55%). Among all clusters "Metal processing" and "High-temperature processes" have the lowest contribution to elemental fine and coarse mass amounting to 1 – 6% and 4 – 9%, respectively, supporting the influence of local emission sources.





### 3.5 Saharan dust outbreaks

Although African dust outbreaks are more frequent in the Mediterranean basin, the long-range transport of dust mainly from
the Sahara desert is not unusual over Poland (e.g., Janicka et al., 2017). Such dust events were observed at the beginning (18
February) and at the end (8 March) of the measurement campaign. The back trajectory analysis shows that during the first
event, the air masses observed over Warsaw at 3 000 m were advected from near-surface heights over North Africa in the
warm sector of an southeastward-moving Mediterranean cyclone, and rose along the warm front while displacing
northeastwards off Africa. In turn, air parcels observed at 1 500 m descended from the upper troposphere behind the surface
cold front of the cyclone, being also directed northeastwards. The trajectories reaching the sampling point at 200 m,
however, came from the east (Fig. 8). The southwesterly airflows were observed for 2 days from February 18, 01:00 until
February 19, 23:00. The inflow from over Sahara desert is also evident during the second event, when long-range transport
of African dust is observed for trajectories reaching Warsaw at 3 000 m (Fig. 8), after mobilization over North Africa and
advection by the warm sector winds on the foreside of a cold front trailing from a developed Genoa Low and primary
cyclone moving in higher latitudes from the Netherlands to southern Germany.

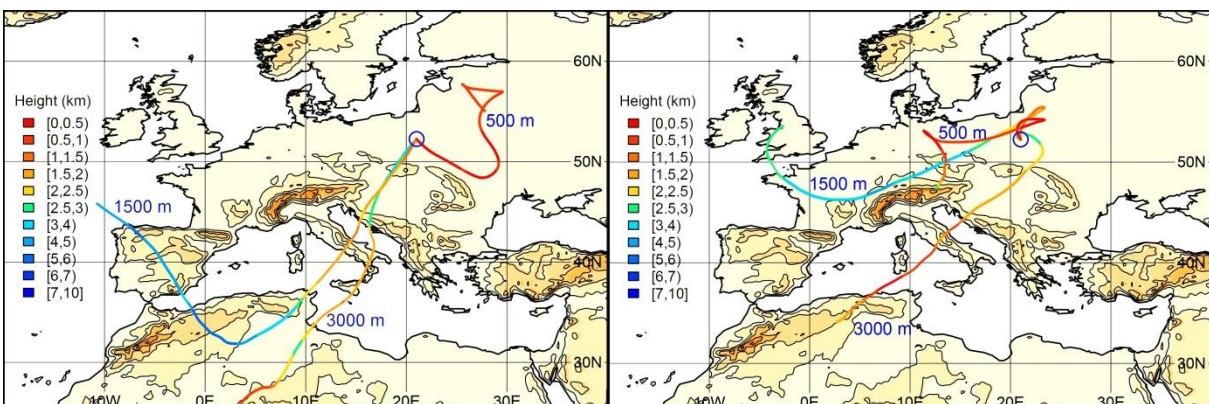

**Figure 8: Air mass trajectories arriving to Warsaw at 200, 1 500 and 3 000 m a.s.l. at 08:00 UTC on 18 February (left panel) and at
12:00 UTC on 8 March (right panel). The colors along a trajectory indicate the height of the air parcels.**

The measurements of many elemental species allowed for estimation of the concentrations of soil dust, being a significant
PM component. The soil dust component was calculated based on the concentrations of the oxides of the five major
elements, i.e.: Al, Si, Ca, Ti and Fe, following a wide used approach given by Eq. (8) (e.g., Chow et al., 2015):

$$Soil\ dust = 2.20 \cdot [Al] + 2.49 \cdot [Si] + 1.63 \cdot [Ca] + 1.94 \cdot [Ti] + 2.42 \cdot [Fe] ,\qquad\qquad (8)$$

As can be seen in Fig. 9, strong soil dust component peaks may be observed during the days with air masses coming from
Africa. In the fine fraction, the recorded peak values are as high as 2.0 µg m$^{-3}$ and 3.5 µg m$^{-3}$ during the first and second
episode, respectively. In the coarse fraction, the concentrations of the estimated soil dust are higher with the peak levels
equal 4.7 µg m$^{-3}$ and 6.1 µg m$^{-3}$ during the first and second episode, respectively. In comparison, the mean concentrations of

soil dust during non-episode days equaled 0.5 µg m⁻³ and 1.1 µg m⁻³ in the fine and coarse fraction, respectively. In both fractions, the time series of "Soil dust" sources identified by the PMF show the same peak values and their concentrations

are highly correlated with those of the calculated soil dust ($r = 0.75$ and $r = 0.85$ for the fine and coarse fraction, respectively), thus strengthening the attribution of these PMF factors to the soil dust.

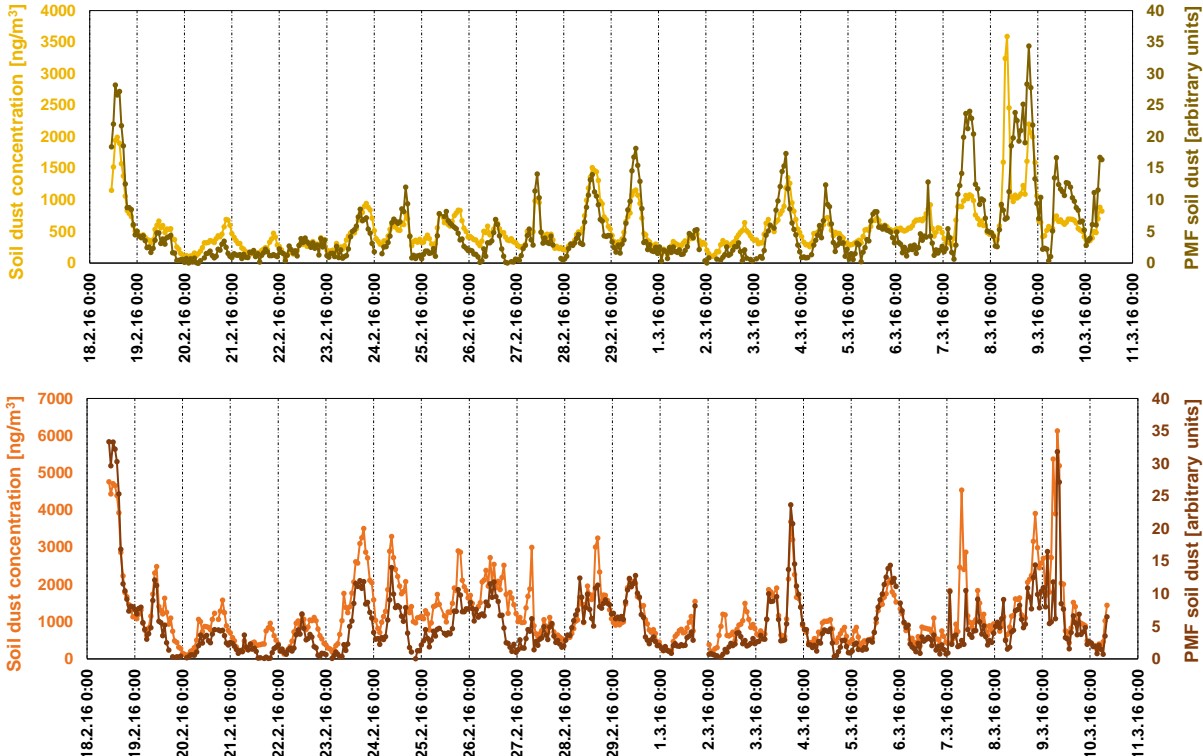

**Figure 9: Hourly time series of the "Soil dust" source contributions identified by PMF analysis (dark colors) and calculated soil**

**component concentrations (light colors) in the fine (upper panel) and coarse (bottom panel) fractions.**

## 4 Summary and Conclusions

The analysis of the composition of trace elements in the fine and coarse fractions of particulate matter at an urban background site in central Warsaw during a high time resolution wintertime measurement campaign has been carried out for the first time in Central Europe.

The source apportionment analysis by the means of 3 receptor models, including multivariate (PMF), wind- (CPF) and trajectory-based (cluster analysis) RMs, allowed for identification of 7 factors for the fine fraction and 5 factors for the coarse one. Traffic-related sources (soil dust mixed with road dust, road dust, exhaust and non-exhaust traffic emissions) had the biggest contribution in the coarse elemental mass (together accounting for 83%), followed by sulfates from local emissions (11.5%). In the fine fraction, regionally transported aged secondary sulfate was found as the major source (44%),



followed by traffic-related sources (20%). Such high contribution of transport of secondary sulfates was encouraged by the unusual for the Central European urban area content of secondary inorganic aerosols (35%) measured in $PM_{2.5}$ mass during the measurement campaign. The share of remaining sources in both fractions did not exceed 15%, with the lowest contribution of industry-related sources (metal processing and high-temperature processes) accounting for 2% and 6% in the fine and coarse fraction, respectively. The source polar plots based on the hourly wind data supported the identification of

the possible origin areas of different sources identified by PMF.

We can conclude that presented findings are consistent with the previous study performed at the same urban background site (Juda-Rezler et al., 2020), which demonstrated residential sector and road transport as the predominant sources for $PM_{2.5}$ pollution in Warsaw, using daily concentrations of $PM_{2.5}$ and its constituents, i.e., 8 ions, carbonaceous matter (EC, OC) and 21 trace elements. However, exploiting of high time resolution elemental data, despite the lack of macro components (i.e.,

ions, carbonaceous components), improves the source apportionment modelling results by allowing the identification of sources that were not identified with daily time resolution, such as wood combustion and Cl-rich source. Moreover, only high time resolution of the concentrations of elements makes it possible to reveal and conclude about "multi-faces" of a given element, which may have different sources in different fractions of PM. Such features were found for Cl and Cr, for which distinct temporal profiles and correlation with different elements allowed for identification of diverse origin of these

elements in the fine and coarse fraction. Cl was apportioned to the wintertime application of road salt for deicing purposes and mixed source of particles rich in Cl in the coarse and fine fraction, respectively, while Cr demonstrated an industrial origin in fine fraction and traffic one in the coarse fraction.

Applied trajectory-based receptor model based on the high time resolution is allowing for the following of the transport pathways of the aerosols. Our results showed substantial contribution of regionally transported aged aerosols, in particular

from South and East of Warsaw, while more local impact of the rest of identified sources has been found. Thus, both emission control measures in the upwind areas, as well as control of local emissions (mainly in residential combustion, traffic- and industry-related sectors) should be applied in the city for the effective air pollution control in winter.

The analyses carried out in this study allowed also for identification of two intense Saharan dust outbreaks in Warsaw. During these episodes long-range transport of dust from Sahara desert was observed for trajectories arriving at 1 500 m and 3

000 m. The calculated concentrations of soil dust showed strong impact in both fractions – the levels of soil component during Saharan episodes were 7 and 6 times higher than the mean concentrations observed during non-episodes days in the fine and coarse fraction, respectively.

**Data availability**

The data used in this study will be published in an open data repository, as well as will be available from the corresponding

author upon request.





**Author contributions**

M.R., K.M. and K.J.-R. designed the study. M.R., G.C. and K.M. carried out measurements. G.C., L.C. and F.L. performed the laboratory analysis of the collected samples. J.A.G.O. performed the back-trajectory simulations. M.R. carried out receptor modelling and data visualization, with contribution of all co-authors to the interpretation of the data. The manuscript was written by M.R. with substantial contribution of K.J-R. and all co-authors. K.J.-R. was responsible for supervision and funding acquisition in Polish National Science Centre.

**Competing interests**

The authors declare that they have no conflict of interest.

**Acknowledgements**

The authors gratefully acknowledge the Municipal Water Supply and Sewerage Company (MPWiK) in Warsaw for the help in organizing the measurement campaign.

**Financial support**

This work was supported by the Polish National Science Centre (Narodowe Centrum Nauki) under OPUS funding scheme 7th edition, Project no. UMO-2014/13/B/ST10/01096.

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
