# Peer review of "Measurement report: Receptor modeling for source identification of urban fine and coarse particulate matter using hourly elemental composition"

_Atmospheric Chemistry and Physics, 2021_

## Author Comment (AC1)

**Manuscript Number: acp-2021-253**

**Authors: Magdalena Reizer, Giulia Calzolai, Katarzyna Maciejewska, José A. G. Orza, Luca Carraresi, Franco Lucarelli, Katarzyna Juda-Rezler**

**Title: Measurement report: Receptor modelling for source identification of urban fine and coarse particulate matter using hourly elemental composition**

We would like to thank the Anonymous Reviewer #1 for the assessment of our manuscript and for sound and constructive comments. The authors appreciate a lot the work that Reviewer put to help us in improving our paper. We took into account comments and suggestions of the Reviewer, and performed revision of the manuscript, trying to clarify all issues. The Reviewer's comments are in italics; our responses are in dark blue.

**Response to the comments of Anonymous Referee #1 (22 Apr 2021)**

*General comments*

*The manuscript presents one month of hourly measurements of major and trace elements in PM2.5 and PM10 in Warsaw, Poland, in February and March 2016. The data are thoroughly discussed, and three different receptor models are applied to determine the sources and origins of the elements. Five sources in PM10 and seven sources in PM2.5 are found, demonstrating the advantages of high time resolution for appropriate source identification. Furthermore two cases of Saharan dust transport are discussed..*

*The structure of the manuscript, the results and the presentation of the material are very detailed and carefully worked out. The topic is relevant and well worth publication in ACP. I would, however, suggest a few minor changes and additions before publication.*

*Specific comments*

*The advantage of +/- hourly time resolution of elemental concentrations is nicely demonstrated, but biased towards Streaker sampling and PIXE analysis. Recent studies with XRF method have also achieved hourly resolution and size segregation for source apportionment, e.g. see references for Beijing and Delhi in Rai et al. (2021):*

*Rai, P., Slowik, J. G., Furger, M., El Haddad, I., Visser, S., Tong, Y., Singh, A., Wehrle, G., Kumar, V., Tobler, A. K., Bhattu, D., Wang, L., Ganguly, D., Rastogi, N., Huang, R. J., Necki, J., Cao, J., Tripathi, S. N., Baltensperger, U., and Prévôt, A. S. H.: Highly time-resolved measurements of element concentrations in PM10 and PM2.5: comparison of Delhi, Beijing, London, and Krakow, Atmos. Chem. Phys., 21, 717-730, 10.5194/acp-21-717-2021, 2021.*

Thank you for this comment. We have added in the manuscript more references showing the use of hourly sampling as well as a statement clarifying that the examples are referred to both the streaker sampler and Xact metals monitor. This paragraph (lines 64-80) has been modified as follows (changes to the manuscript are indicated in red font):

"Only limited number of studies have applied receptor modelling techniques to the high-time resolution PM samples, particularly applying Positive Matrix Factorization (PMF) model to the hourly elemental composition of $PM_{2.5}$ and $PM_{2.5-10}$ samples. There are only few measurement devices allowing for the sampling of the concentrations of the elements with high time resolution, with the wide application of the streaker sampler (PIXE International Corporation) (Calzolai et al., 2015) and the semi-continuous X-ray fluorescence spectrometer Xact Ambient Metals Monitor (Cooper Environmental Services) (Rai et al., 2020). Identification of PM sources based on aerosol samples in 1-hour resolution has been carried out in Southern Europe,

e.g.: at 4 urban sites in Barcelona (Spain), Porto (Portugal), Athens (Greece) and Florence (Italy) (Lucarelli et al., 2015); at an urban site in Elche (Spain) (Nicolás et al., 2020); at 6 sites of different types in Tuscany (Central Italy) (Nava et al., 2015), in a small town in the Po Valley (Italy) (Belis et al., 2019); and in an industrial area of Taranto (Italy) (Lucarelli et al., 2020). Outside this region, hourly-resolved PM samples have been investigated e.g.: in London (Crilley et al., 2017) in the United Kingdom and at 4 different sites in Alexandra in New Zealand (Ancelet et al., 2014). There has been also the wide application of both measurement devices in Asia, e.g.: in Wuhan (Acciai et al., 2017), the Pearl River Delta region (Zhou et al., 2018), Shanghai (Chang et al., 2018) and Beijing (Rai et al., 2021) in China; as well as in the capital of India, Delhi (Rai et al., 2020). However, according to our knowledge, receptor modelling studies based on hourly elemental composition of PM has not been carried out in Central Europe previously".

The following references has been added:

Acciai, C., Zhang, Z., Wang, F., Zhong, Z., and Lonati, G.: Characteristics and source analysis of trace elements in PM2.5 in the urban atmosphere of Wuhan in spring, Aerosol Air Qual. Res., 17, 2224–2234, doi: 10.4209/aaqr.2017.06.0207, 2017.

Chang, Y., Huang, K., Xie, M., Deng, C., Zou, Z., Liu, S., and Zhang Y.: First long-term and near real-time measurement of trace elements in China's urban atmosphere: temporal variability, source apportionment and precipitation effect, Atmos. Chem. Phys., 18, 11793–11812, doi:10.5194/acp-18-11793-2018, 2018.

Rai, P., Furger, M., El Haddad, I., Kumar, V., Wang, L., Singh, A., Dixit, K., Bhattu, D., Petit, J. -E., Ganguly, D., Rastogi, N., Baltensperger, U., Tripathi, S. N., Slowik, J. G., and Prévôt, A. S. H.: Real-time measurement and source apportionment of elements in Delhi's atmosphere, Sci. Total Environ., 742, 140332, doi:10.1016/j.scitotenv.2020.140332, 2020.

Rai, P., Furger, M., Slowik, J. G., Zhong, H., Tong, Y., Wang, L., Duan, J., Gu, Y., Qi, L., Huang, R. -J., Cao, J., Baltensperger, U., and Prévôt, A. S. H.: Characteristics and sources of hourly elements in PM10 and PM2.5 during wintertime in Beijing, Environ. Pollut., 278, 116865, doi:10.1016/j.envpol.2021.116865, 2021.

*In Figs. 4 and 5 the right hand axes are labelled as 'contribution [%]', while in the captions you call this 'explained variation'. Would it not be more consistent (and more correct) to just use 'contribution' in both places?*

The captions of both figures have been changed as follows:

Figure 4: Left panel: PMF profiles (bars, left y axis) and contributions (black diamonds, right y axis) of the identified sources for the fine fraction. Right panel: Daily patterns of the identified sources (in arbitrary units).

Figure 5: Left panel: PMF profiles (bars, left y axis) and contributions (black diamonds, right y axis) of the identified sources for the coarse fraction. Right panel: Daily patterns of the identified sources (in arbitrary units).

*In Fig. 6 the traffic source area is quite different from the road salt source. Shouldn't we expect more similarity between the two factors, as in both cases probably resuspension would be the mechanism for 'creating' the sources? Or is deicing salt not evenly distributed along the Warsaw road system? Please discuss this a bit more.*

Thank you for this comment. In the case of "Road salt" source, the origin area is located in the part of Warsaw where main road from the city's airport and one of the main expressway running through Warsaw are located. This road could be intensively de-iced during the measurement campaign. Moreover de-icing activities at the airport could also influence the concentrations as sodium-based compound are used for the de-icing of airports' runways and roads. Also daily profiles of this source suggest the resuspension processes. In contrary, "Traffic" source identified for both fine and coarse fractions shows bi-modal daily profiles characteristic for the traffic sources. The CPF analyses show different location than for "Road salt" sources. The measurement site was located in the city center where exhaust emission is probably playing a key role.

*Fig. 8 shows trajectories originating in or crossing over parts of the Sahara desert and ending above Warsaw in 1500 and 3000 m asl. It is not straight forward that PM arriving at these elevations is measured with ground-based samplers, and possible downward mixing processes should be discussed in more detail. This is especially the case on 18 Feb, where at 500 m asl the airmass is advected from SE, indicating a completely different source location than the Sahara. While I find these two cases interesting and plausible, a strong connection is not given. I recommend adding one or two sentences discussing the uncertainties (Saharan dust composition, vertical mixing from upper layers, concentrations aloft).*

Thank you for this comment. Following the recommendation we provide additional support to the occurrence of the African dust outbreak of February 2016 over Warsaw. We have included several plots as Supplementary Material: the MODIS true color image for 15 February, the output of the DREAM prediction model for 18 February and the radiosounding launched near Warsaw on 18 February. Cloud cover prevented from having optical remote sensing measures that would provide information regarding the vertical load/distribution of dust in this episode. In addition, we have included in the main text (subsection 3.5) a more detailed description that allows relating the presence of dust at low levels with the height of the trajectories carrying African dust.

"The advection of dust-laden airflows out of Africa toward Warsaw is evidenced by satellite imagery on 15 February (Fig. S24), and both back trajectories (Fig. 8) and dust prediction models (Fig. S25) indicate they reach the study area aloft by 18 February. However, the small-scale downward movement and mixing of dust particles is not represented in the back trajectory model and therefore it cannot explain the dust impact at the ground level. Besides, cloud coverage prevented optical remote sensing monitoring. Therefore the presence of dust at low levels is supported primarily by the $PM_{2.5}$ chemical composition and the PMF analysis as shown below. The operational radiosounding launched on 18 February in Legionowo (12374), around 20 km to the north of Warsaw, and surface meteorological instruments confirm the near-surface southeasterly winds in association with a high-pressure system located to the northeast of Poland, consistent with the trajectories found at 200 m a.s.l. Half-hourly wind speed at the Warsaw Okęcie Airport was over 20 km h$^{-1}$ only from 00:00 UTC to 02:30 UTC, so the $PM_{2.5}$ soil contribution registered on 18 February (Fig. 9) is unlikely due to local/regional soil particulate resuspension. The radiosounding (Fig. S26) shows the presence of a dry layer from near 1 500 m asl to over 2 500 m asl where wind have veered to southerly. This layer overlaps the top of a temperature inversion layer with base (the atmospheric boundary layer) at 750 m a.s.l., located in a cloud top. Above the dry layer, a thick ice cloud extends above 5000 m a.s.l. The trajectories corresponding to the dry layer have a North African origin; therefore it can be identified as a dust layer. As the dust layer overlaps the top of the inversion layer and the

entrainment zone is at the base of the inversion, near the cloud top, the dust layer is then near the entrainment zone making possible the mixing of dust within the ABL down to the ground."

[Figure]

Figure S24. Aqua MODIS corrected reflectance (true color) on 15 February, 2016. NASA EOSDIS Worldview (https://worldview.earthdata.nasa.gov/).

[Figure]

Figure S25. Aerosol Optical Depth (AOD) forecast for 18 February, 12 UTC. Image from the NMMB/BSC-Dust model operated by the Barcelona Supercomputing Center, Spanish National Supercomputing Center (https://www.bsc.es/ess/bsc-dust-daily-forecast/).

[Figure]

Figure S26. Skew T-log p diagrams for Legionowo on 18 February, 12 UTC. University of Wyoming (http://weather.uwyo.edu/upperair/sounding.html).

*In Fig. 9, I do not see an advantage of comparing the elemental time series in ng/m3 with the PMF time series in arbitrary units, as the main elements comprising the soil dust are those in eq. (8), so we basically just add O (in stoichiometric ratios) to these elements. It should be possible from a linear regression to estimate the fraction of the two (I guess it would be something around the value of 2). With respect to the Saharan dust, it would be the same to just compare the five relevant elements, not their oxides, as long as we do not have quantitative information on the amount of dust (mass) transported.*

Thank you for this comment. Our main goal of this part of the manuscript was to support the apportion of the "Soil dust" source identified by PMF. For this purpose, the soil dust component was chemically reconstructed by using the conversion of elemental concentrations of typical crustal species Al, Si, Ca, Ti and Fe following the approach widely reported in the literature (e.g.: Chow et al. (2015)). As these elements are predominantly present in the soil in the oxidized forms this approach assumes the conversion of elements to their oxides.

Chow, J. C., Lowenthal, D. H., Chen, L.-W. A, Wang, X. and Watson, J. G.: Mass reconstruction methods for $PM_{2.5}$: a review, Air Qual. Atmos. Hlth., 8, 243–263, doi:10.1007/s11869-015-0338-3, 2015.

*Technical corrections*

*L118    device (delete s)*

It has been corrected.

*L121    substrata (insert s)*

It has been corrected.

*L126    beam (delete s)*

It has been corrected.

*L131    write Micromatter in one word*

It has been corrected.

*L141    Reorder the sentence. Write '… is a widely used (…) multivariate factor analysis model in air quality studies based on …'*

It has been corrected.

*L142    weighted least squares fit (add an s)*

It has been corrected.

*L369    Can you explain the term 'bioavailable' a bit better? The way you use it, it appears to be something like a quantitative entity with a time dependence.*

Bioavailability is defined as the degree and rate of absorption of a substance by the living systems or the degree which the substance is available to physiologically active sites. The following explanation has been added to the text (changes to the manuscript are indicated in red font):

"It is noteworthy that in wintertime in Warsaw, As and K appeared to be highly bioavailable, indicating the high degree and rate of absorption of a substance by the living systems or the high degree which the substance is available to physiologically active sites. During the days of elevated air pollution levels higher bioavailability of those element was observed, suggesting a higher risk to humans posed by emission from this source during those days (Juda-Rezler et al., 2021)."

*L373    … emitting a substantial amount … (insert a)*

It has been corrected.

*L377    … attributed to wood combustion … (delete the)*

It has been corrected.

*L387    In the case of the coarse …  (insert the)*

It has been corrected.

*L393    … the pattern of the source … (insert the)*

It has been corrected.

*L504    … points towards two …   (insert towards)*

It has been corrected.

*L516    … all ranges of wind … (insert s)*

It has been corrected.

*L518    … are used for the maintenance … (replace to with 'for the')*

It has been corrected.

*L533    write '…of the sources identified by PMF for….'*

It has been corrected.

*L592    widely (instead of wide)*

It has been corrected.

*L633    … receptor modeling based …  (insert ing)*

It has been corrected.

---

## Author Comment (AC2)

**Manuscript Number: acp-2021-253**

**Authors: Magdalena Reizer, Giulia Calzolai, Katarzyna Maciejewska, José A. G. Orza, Luca Carraresi, Franco Lucarelli, Katarzyna Juda-Rezler**

**Title: Measurement report: Receptor modelling for source identification of urban fine and coarse particulate matter using hourly elemental composition**

We would like to thank the Anonymous Reviewer #2 for the assessment of our manuscript and for sound and constructive comments. The authors appreciate a lot the work that Reviewer put to help us in improving our paper. We took into account comments and suggestions of the Reviewer, and performed revision of the manuscript, trying to clarify all issues. The Reviewer's comments are in italics; our responses are in dark blue.

**Response to the comments of Anonymous Referee #2 (21 May 2021)**

*The manuscript presents a study to identify the sources of fine and coarse particulate matter based on high resolution sampling of elemental components in the city of Warsaw. The sample collection and the data treatment were carried out according to state-of-the art methodologies. The manuscript is well organized and complete. The figures are of good quality and summarize the data properly. The discussion of some of the sources needs to be improved as pointed out in the specific comments. The text should be revised by a native speaker.*

Thank you for this comment. The manuscript will be checked by English native speaker before final submission.

*Specific comments*

*Line 81: the year should be between parentheses*

It has been corrected.

*Line 141: Maybe Belis et al., 2020 (https://doi.org/10.1016/j.aeaoa.2019.100053) is a more appropriate citation for this paragraph*

The reference Belis et al. (2019) has been replaced by Belis et al. (2020).

*Line 171: The authors should discuss what's the impact of the absence of such important components on the ability of the analysis to quantify sources*

This paragraph has been modified as follows (changes to the manuscript are indicated in red font):

"In this study, concentrations of PM mass and its macro components, i.e., organic carbon, elemental carbon, secondary inorganic aerosols and other water-soluble inorganic ions, were not available with the hourly temporal resolution. Therefore, the analysis of the hourly concentrations of the elements have been used for the detailed identification of PM sources supplementing the previous source apportionment study performed by Juda-Rezler at al. (2020). Due to the lack of PM mass in the measurement campaign, the contribution of the identified sources to the total aerosol mass cannot be provided and the source time series will be expressed in arbitrary units (see e.g., Lucarelli et al., 2020)".

*Line 222: Please, discuss if the soluble ions are comparable with the elemental determinations*

The following discussion has been added to the text:

"Simultaneously, to the streaker data, daily concentrations of PM$_{2.5}$ and its components were collected. The daily elemental concentrations (averaged from the hourly data) and daily concentrations of the respective water-soluble ions ($SO_4^{2-}$, $K^+$, $Cl^-$, $Na^+$) was compared. A general good agreement was found, with a strong correlation between elemental sulfur and $SO_4^{2-}$ ($r = 0.97$), as well as between elemental and water-soluble potassium ($r = 0.74$). For elemental and water-soluble chlorine and sodium moderate correlations ($r = 0.69$ and $r = 0.61$, respectively) was observed. Missing data and data below the detection limit did not allow the comparison of the concentrations of calcium and magnesium."

*Lines 220 – 228: This paragraph should go to materials and methods*

The paragraph has been moved to "Materials and method" to the chapter 2.2 ("PM sampling and elements determination").

*Lines 249-251: This paragraph should go to materials and methods*

The paragraph has been moved to "Materials and method" to the chapter 2.1 ("Study area").

*Line 369: The bioavailability issue should be explained/discussed more in detail*

The definition of the bioavailability has been added to the text (changes to the manuscript are indicated in red font):

"It is noteworthy that in wintertime in Warsaw, As and K appeared to be highly bioavailable, indicating the high degree and rate of absorption of a substance by the living systems or the high degree which the substance is available to physiologically active sites. During the days of elevated air pollution levels higher bioavailability of those element was observed, suggesting a higher risk to humans posed by emission from this source during those days (Juda-Rezler et al., 2021)."

*Line 396: Although sulfate is a common component of biomass burning profiles, claiming this fuel is a major contributor to secondary sulfate should be better supported by evidence. Especially in an area where coal combustion, a well-known SO2 source, is well documented.*

This paragraph has been modified as follows (changes to the manuscript are indicated in red font):

"This confirms that the fine fraction is dominated by regional rather than local transport of SO$_2$ and sulfate from Warsaw's outskirts with individual residential heating, mainly with the use of coal and biomass, whereas in the coarse fraction the presence of sulfate is probably due to the local combustion activities in the city itself."

*Line 494: Consider calling this source mixed wood and coal combustion*

Thank you for this comment. We agree, the name of the source has been changed to "Wood and coal combustion".

*Line 568: The aged sea salt contribution may be higher than the authors' estimations considering that what is identified as "road salt" has the highest shares in the three clusters originated in the North Atlantic*

This paragraph has been modified as follows (changes to the manuscript are indicated in red font):

"This may suggest that although road salt is the main source of atmospheric aerosol in Warsaw, the partial contribution of sea spray is also highly probable."

*Line 580: I guess you mean "m above ground level" please, specify*

The trajectories arriving to Warsaw were calculated at all 3 heights (i.e. 200 m, 1 500 m, 3 000 m) above sea level. This sentence has been corrected:

"The trajectories reaching the sampling point at 200 m a.s.l., however, came from the east (Fig. 8)."

*Lines 615-617: This sentence is not clear, please, rephrase*

The sentence has been changed as follows (changes to the manuscript are indicated in red font):

"Such high contribution of transport of secondary sulfates was also found in the daily PM$_{2.5}$ data showing unusual for the Central European urban area content of secondary inorganic aerosols (35%) during the measurement campaign."

*Line 622: excluding the secondary aerosol*

In the study Juda-Rezler et al. (2020) the source representing residential sector was identified as "Residential combustion (fresh and aged aerosol)", including PM primary emitted within the city as well as aged aerosol transported from the outskirts of the city. Therefore, this sentence (lines 621-624) has been modified as follows (changes to the manuscript are indicated in red font):

"We can conclude that presented findings are consistent with the previous study performed at the same urban background site (Juda-Rezler et al., 2020), which demonstrated combustion in the residential sector within the city and in the surrounding suburban areas, followed by road transport as the predominant sources for PM$_{2.5}$ pollution in Warsaw, using daily concentrations of PM$_{2.5}$ and its constituents, i.e., 8 ions, carbonaceous matter (EC, OC) and 21 trace elements".

*Table 1: you should point out when min values are below the detection limit here*

The minimum concentrations below the detection limits have been indicated by the asterisks, as follows:

**Table 1: Descriptive statistics for the hourly concentrations (ng m⁻³) of the elements measured in fine and coarse fractions. Minimum concentrations below the detection limit are indicated by an asterisk.**

| Element [ng m⁻³] | Fine | | | | | Coarse | | | | |
|---|---|---|---|---|---|---|---|---|---|---|
| | Mean | Median | Min | Max | SD | Mean | Median | Min | Max | SD |
| Al | 25.3 | 18.9 | 3.9* | 206.0 | 24.1 | 70.2 | 50.9 | 2.2* | 477.6 | 64.9 |
| As | 0.5 | 0.3 | 0.1* | 2.3 | 0.4 | 0.2 | 0.1 | 0.04* | 0.9 | 0.1 |
| Ba | 7.8 | 6.5 | 5.1* | 28.1 | 3.7 | 6.5 | 3.4 | 2.1* | 42.9 | 5.4 |
| Br | 2.4 | 2.2 | 0.4 | 9.7 | 1.3 | 0.2 | 0.1 | 0.05* | 3.0 | 0.2 |
| Ca | 48.8 | 31.5 | 4.1* | 287.2 | 45.4 | 108.1 | 84.7 | 4.3 | 449.8 | 88.8 |
| Cl | 113.3 | 60.1 | 5.5* | 1 492.9 | 168.1 | 141.8 | 35.5 | 1.5* | 1 682.5 | 251.3 |
| Cr | 2.5 | 2.3 | 0.1* | 14.4 | 1.3 | 1.3 | 1.1 | 0.2* | 6.9 | 0.9 |
| Cu | 6.7 | 4.5 | 0.7 | 190.2 | 13.2 | 4.9 | 3.6 | 0.2 | 41.3 | 4.8 |
| Fe | 114.7 | 95.4 | 13.2 | 1 296.9 | 105.1 | 193.1 | 154.3 | 11.1 | 1 281.7 | 148.8 |
| K | 166.8 | 153.2 | 18.1 | 639.6 | 85.1 | 29.9 | 24.5 | 2.1* | 152.9 | 20.1 |
| Mg | 17.4 | 17.0 | 4.9* | 48.4 | 8.6 | 36.8 | 25.4 | 3.0* | 207.3 | 33.5 |
| Mn | 2.4 | 2.1 | 0.4* | 21.1 | 1.9 | 2.3 | 2.0 | 0.2* | 11.8 | 1.6 |
| Mo | 0.4 | 0.4 | 0.2* | 2.1 | 0.2 | 0.2 | 0.2 | 0.1* | 0.9 | 0.1 |
| Na | 80.4 | 57.2 | 8.6* | 1 538.5 | 91.8 | 170.6 | 71.7 | 3.5* | 1 347.8 | 231.5 |
| Ni | 0.8 | 0.7 | 0.02* | 8.2 | 0.6 | 0.3 | 0.2 | 0.2 | 2.4 | 0.2 |
| P | 20.9 | 19.8 | 1.8* | 48.7 | 8.3 | 7.0 | 6.5 | 1.3* | 21.7 | 3.3 |
| Pb | 11.7 | 9.9 | 0.5* | 146.2 | 11.5 | 1.0 | 0.3 | 0.2* | 20.5 | 1.5 |
| Rb | 0.3 | 0.2 | 0.1* | 3.1 | 0.2 | 0.2 | 0.1 | 0.1* | 10.7 | 0.5 |
| S | 1 020.8 | 907.6 | 185.2 | 2 612.8 | 548.0 | 73.9 | 55.7 | 6.1 | 468.0 | 62.2 |
| Se | 0.4 | 0.4 | 0.1* | 1.2 | 0.2 | 0.1 | 0.1 | 0.05* | 0.5 | 0.04 |
| Si | 51.4 | 51.4 | 0.7* | 349.1 | 39.2 | 166.1 | 126.4 | 8.9 | 995.5 | 134.4 |
| Sr | 0.5 | 0.3 | 0.2* | 15.8 | 0.9 | 0.4 | 0.3 | 0.1* | 3.1 | 0.4 |
| Ti | 2.9 | 1.7 | 1.3* | 20.6 | 2.4 | 5.7 | 4.4 | 0.6* | 34.6 | 4.8 |
| V | 1.2 | 1.0 | 0.8* | 3.3 | 0.5 | 0.6 | 0.4 | 0.3* | 2.5 | 0.3 |
| Y | 0.3 | 0.3 | 0.2* | 2.3 | 0.1 | 0.1 | 0.1 | 0.1* | 1.1 | 0.1 |
| Zn | 38.7 | 33.0 | 6.7 | 239.5 | 25.3 | 6.5 | 4.5 | 0.4 | 52.2 | 6.0 |
| Zr | 0.4 | 0.3 | 0.2* | 2.2 | 0.2 | 0.4 | 0.2 | 0.1* | 3.3 | 0.4 |

*Figures 4 and 5: Please, put the labels of the x axis categories also at the top for a better visualization.*

The change has been done. As an example the new Figure 5 is presented below. Please note that according to the next comment the colors corresponding to the sources in Figure 5 has been also changed as they were analogous in Figures 4, 5 and 7.

[Figure]

**Figure 1: Left panel: PMF profiles (bars, left y axis) and contributions (black diamonds, right y axis) of the identified sources for the coarse fraction. Right panel: Daily patterns of the identified sources (in arbitrary units).**

*Figure 7: I suggest to use the same colour for corresponding sources in the fine and coarse fraction.*

The colors has been changed as presented in the new Figure 7 below. Please note that also colors corresponding to the sources in Figure 5 has been also changed as they were analogous in Figures 4, 5 and 7.

[Figure]

**Figure 2: Trajectory cluster centroids arriving to Warsaw at 200 m (center map) with PMF factor contribution in different clusters for fine (left pie charts) and coarse (right pie charts) fraction. Percentage of the trajectories classified into each cluster is given in the parentheses.**

---

## Author Response (AR1)

**Manuscript Number: acp-2021-253**

**Authors: Magdalena Reizer, Giulia Calzolai, Katarzyna Maciejewska, José A. G. Orza, Luca Carraresi, Franco Lucarelli, Katarzyna Juda-Rezler**

**Title: Measurement report: Receptor modelling for source identification of urban fine and coarse particulate matter using hourly elemental composition**

We would like to thank the Anonymous Reviewer #3 for the assessment of our manuscript and for sound and constructive comments. The authors appreciate a lot the work that Reviewer put to help us in improving our paper. We took into account comments and suggestions of the Reviewer, and performed revision of the manuscript, trying to clarify all issues. The Reviewer's comments are in italics; our responses are in dark blue.

**Response to the comments of Anonymous Referee #3 (16 June 2021)**

*Authors present results of high time resolution measurement of elements in fine and coarse urban aerosols with subsequent identification of sources using a combination of three receptor models.*

*The paper provides interesting results with detailed insights into the winter pollution sources in Warsaw area.*

*The paper is suitable for publication in the journal Atmospheric Chemistry and Physics, however, there are a few issues that need to be addressed before acceptance for publication. Minor revisions of the paper taking into consideration the comments reported below are requested.*

*Comments:*

*line 19: Exhaust traffic emissions are mostly prevailing in fine (and especially in submicrometre) fraction.*

Traffic-related sources identified in the coarse fraction include soil dust mixed with road dust, road dust, as well as exhaust and non-exhaust traffic emissions. The latter emission sources were identified as more general source "Traffic" as elements characteristic for exhaust and non-exhaust emissions were presented in the PMF profile and detailed separation of both types of emission were not possible. However, most probably the share of non-exhaust emission is prevailing in this fraction, but our study is not able to confirm this.

The abstract has been modified as follows (changes to the manuscript are indicated in red font):

"In the fine fraction, aged sulfate aerosol related with emissions from solid fuel combustion in the residential sector located outside the city was the largest contributing source to fine elemental mass (44%), while traffic-related sources, including soil dust mixed with road dust, road dust, as well as  traffic emissions, had the biggest contribution in the coarse elemental mass (together accounting for 83%)."

*line 34-35: Statement that the greatest health risk is from PM25 is relative. It depends on considered particle size. The statement is true if you compare PM10 and PM2.5. But it is not true for the comparison of PM2.5 and PM1. PM1 and especially UFP are more dangerous than PM2.5 due to their ability to penetrate deeper into the lung than PM2.5 particles.*

Thank you for this comment. We have focused on the $PM_{10}$ and $PM_{2.5}$ as those fractions are commonly studied in the cohort studies worldwide. This paragraph has been modified as follows (changes to the manuscript are indicated in red font):

"Many epidemiological studies have shown strong relationship between PM and adverse health effects, focusing on either short-term or long-term exposure (e.g., Pope and Dockery, 2006; Pope et al., 2018). Majority of the worldwide cohort studies used $PM_{10}$ and/or $PM_{2.5}$ (particles with aerodynamic diameter smaller than 10 μm and 2.5 μm, respectively) as the exposure metric. Comparing these two main fractions of PM, the greater risk to health is posed by $PM_{2.5}$ (particles with aerodynamic diameter smaller than 2.5 μm), as it can penetrate the respiratory system via inhalation, causing or aggravating respiratory and cardiovascular diseases, reproductive and central nervous system dysfunctions, as well as cancer (e.g., Manisalidis et al., 2020). Globally, ambient $PM_{2.5}$ air pollution contributed to 4.14 million deaths in 2019 (Murray et al., 2020)."

*line 73: It is not true, you have overlooked some papers, for example, Pokorná et al., Sci. Total Environ. 2015, 502, 172–183.*

Thank you for this comment. We have focused on the source apportionment using PMF model applied to the hourly elemental composition of $PM_{10}$, $PM_{2.5-10}$ and/or $PM_{2.5}$. In the given reference (Pokorná et al., 2015) the coarse and fine fractions were defined in different way, as $PM_{0.15-1.15}$ and $PM_{1.15-10}$, respectively. Thus, this reference was not included in the paper. However, this paragraph has been clarified as follows (changes to the manuscript are indicated in red font):

"However, according to our knowledge, receptor modelling studies based on hourly elemental composition of $PM_{2.5}$ and $PM_{2.5-10}$ has not been carried out in Central Europe previously."

*line 118: Add sampling flow rate and volume of passed air per sample.*

The information has been added (changes to the manuscript are indicated in red font):

"The aerosol was collected by a sampling device (PIXE International Corporation (Calzolai et al., 2015)) designed to separate the fine (<2.5 μm aerodynamic diameter) and the coarse (2.5–10 μm) modes of atmospheric aerosol at an air flow rate of $1 \, l \, min^{-1}$."

*line 281: Cl originates also from various combustion sources, more details see for example in Mikuška et al, Atmosphere 2020, 11, 688.*

We agree that sea/road salt is not the only source of Cl emission. However, no or weak correlation with other elements and no diurnal pattern of Cl concentrations do not allow identifying any particular source of this element in the fine fraction. Only moderate correlation with Br and K, may suggest the combustion processes as a source, but other analyses performed in the study do not confirm this. The information on Cl sources has been added (changes to the manuscript are indicated in red font):

"Cl is usually attributed either to the sea salt in areas close to the coasts or to the road salt in continental areas of Central and Northern Europe (Belis et al., 2013). It is also emitted from the

combustion of coal, wood and solid waste, in particular in residential sector (e.g., Mikuška et al., 2020).  Recorded time series of Cl in Warsaw are different in the fine and coarse fractions with no correlation between concentrations in the two modes (r = 0.08)."

Mikuška, P., Vojtěšek, M., Křůmal, K., Mikušková-Čampulová, M., Michálek, J., and Věcěra, Z.: Characterization and source identification of elements and water-soluble ions in submicrometre aerosols in Brno and Šlapanice (Czech Republic), Atmosphere-Basel, 11(7), 688, doi: 10.3390/atmos11070688, 2020.

*line 363-384: Component contribution and time profile of this factor in coarse fraction suggest considering renaming this factor to residential heating.*

Sources in both fractions are connected with residential sector as was explained in the text (Chapter 3.3.1). Both sources has similar chemical profile, with sulfur as the main component and thus they were named in similar way. However, in order to differentiate these sources in both fractions, we have used more precise names as: "Aged secondary sulfate" and "Local sulfate" for the fine and coarse fraction, respectively.

*line 466-469: Mentioned metals could also indicate emissions from waste incinerator. Is there any incinerator in the vicinity?*

There is a small municipal waste incinerator in Warsaw, however it is located north-east of the measurement point and cannot influence the measurement point what was confirmed by the CFP analyses. Moreover, lack of Cl in the profile of the sources identified in both fractions do not suggest that sources in both fractions can be related with emissions from waste incinerator.

*line 495-501: Br, Se, As are strong markers of coal combustion, so this factor looks rather like combined combustion of coal and biomass (wood).*

This issue was also raised by Reviewer #2. The name of the source has been changed to "Wood and coal combustion".

*line 608-609: As far as I know other studies are providing high time resolution measurement in Central Europe, see comment for line 73.*

As was stated in the response to the comment for line 73, our analyses and literature review have been focused on the PMF source apportionment applied to the hourly elemental composition of $PM_{10}$, $PM_{2.5-10}$ and/or $PM_{2.5}$. The studies with different definition of fine and coarse fraction as well as studies using 2-h time resolution have been excluded. This paragraph has been clarified as follows (changes to the manuscript are indicated in red font):

"The analysis of the composition of trace elements in the fine ($PM_{2.5}$) and coarse ($PM_{2.5-10}$) fractions of particulate matter at an urban background site in central Warsaw during a high time resolution wintertime measurement campaign has been carried out for the first time in Central Europe."

*line 626-627: According to my experience, parallel measurement of different PM fractions with shorter time resolution can also provide the same results as high time resolution measurement.*

Thank you for this comment.

---

## Author Response (AR2)

**Manuscript Number: acp-2021-253**

**Authors: Magdalena Reizer, Giulia Calzolai, Katarzyna Maciejewska, José A. G. Orza, Luca Carraresi, Franco Lucarelli, Katarzyna Juda-Rezler**

**Title: Measurement report: Receptor modelling for source identification of urban fine and coarse particulate matter using hourly elemental composition**

We would like to thank the Editor for very careful review of our paper, and for the comments and corrections. We took into account all comments in the revised version of the manuscript

**Editor comments**:

*The authors have reasonably addressed the comments of the three anonymous referees and they have modified their manuscript accordingly. However, many alterations and corrections are needed for the Main text before the manuscript can be published in ACP:*

*Line 14: Replace "models, was" by "models was".*

*Line 20: Replace "of identified" by "of the identified".*

*Line 26: Replace "of different" by "of the different".*

*Line 33: Replace "show strong" by "show a strong".*

*Line 34: Replace "Majority" by "The majority".*

*Line 60: Replace "in residential" by "in the residential".*

*Line 61: Replace ", hourly" by ", a hourly".*

*Line 65: Replace "Only limited" by "Only a limited".*

*Line 71: Replace ", following" by ", the following".*

*Line 86: Replace "of urban" by "of the urban" and replace "important ," by "important,".*

*Line 91: Abbreviations and acronyms, here "SA", should be defined (written full-out) when first used. Since "SA" does not occur further in the manuscript. it should be replaced here by "source apportionment".*

*Line 135: Replace "PIXE" by "The PIXE".*

*Line 138: Replace "PIXE" by "The PIXE".*

*Line 152: Replace ", using" by ", using the".*

*Line 168: Replace "for measurement" by "for the measurement".*

*Line 185: Replace "with measured" by "with the measured" and replace "for fine" by "for the fine".*

*Line 186: Replace "coarse" by "the coarse".*

*Line 187: Replace "revealed stable" by "revealed a stable".*

*Lines 196-197: Replace "The bivariate" by "Bivariate".*

*Line 213: Replace "Trajectories" by "The trajectories".*

*Line 224: The "a" before "cluster" should not be in italic.*

*Line 256: Replace "26% of" by "26% of the".*

*Line 257: Replace "comparing to" by "compared to".*

*Line 258: Replace "of fine" by "of the fine".*

*Line 269: Replace "in fine" by "in the fine".*

*Line 272: Replace "of fine" by "of the fine".*

*Line 275: Replace "analyzed" by "measured".*

*Line 279: Replace "Different" by "A different".*

*Line 285: Replace "of concentration" by "of the concentration".*

*Lines 298-299: Replace "in residential" by "in the residential".*

*Line 300: Replace "between concentrations" by "between the concentrations".*

*Line 303: Replace "Concentrations" by "The concentrations".*

*Line 330: Replace "in residential" by "in the residential".*

*Line 331: Replace "and moderate" by "and a moderate".*

*Line 333: Replace "fraction," by "fraction;".*

*Line 337: Replace "2.5-times" by "2.5 times".*

*Line 341: Replace "also similar" by "also a similar".*

*Line 343: Replace "suggest crustal" by "suggests a crustal".*

*Line 345: Replace "As it was described" by "As described".*

*Line 346: Replace "with daily" by "with the daily".*

*Line 382: Replace "exhibits similar" by "exhibits a similar".*

*Line 387: Replace "levels higher" by "levels a higher".*

*Line 393: Replace "Also higher" by "Also a higher".*

*Line 399: Replace "and up to" by "to up to".*

*Line 407: Replace "with water" by "with the water".*

*Line 426: Replace "and notable contribution" by "and a notable contributor".*

*Line 427: Replace "typically crustal" by "typical crustal".*

*Line 432: Replace "mixed source" by "a mixed source" and replace "with substantial" by "with a substantial".*

*Line 433: Replace "Diurnal" by "The diurnal".*

*Line 440: Replace "and noticeable" by "and a noticeable".*

*Line 442: Replace "MgCl" by "MgCl2".*

*Line 457: Replace "of coarse" by "of the coarse".*

*Line 460: Replace "of Fe" by "of the Fe".*

*Line 473: Replace "strongly" by "strong".*

*Line 483: Replace "of Cu" by "of the Cu".*

*Line 488: Replace "in power" by "in the power".*

*Line 490: Replace "the elevated concentrations occurring atnight" by "elevated concentrations occurring at night".*

*Line 491: Replace "exhibited" by "observed".*

*Line 497: Replace "placed" by "situated".*

*Line 512: Replace "of Zn" by "of the Zn".*

*Line 516: Replace "in power" by "in the power".*

*Line 521: Replace "levels from" by "levels from the".*

*Line 526: Replace "in residential" by "in the residential".*

*Line 530: Replace "of Br" by "of the Br".*

*Line 533: Replace "where substantial" by "where a substantial".*

*Line 534: Replace "are used" by "is used".*

*Line 543: Replace "by EPA" by "by the EPA".*

*Line 545: Replace "of base" by "of the base".*

*Line 549: Replace "for fine" by "for the fine".*

*Line 554: Replace "for fine" by "for the fine" and replace "Percentage" by "The percentage".*

*Line 555: Replace "in the parentheses" by "in parentheses".*

*Line 565: Replace "which starts" by "which start".*

*Line 566: Replace "composed of the" by "composed of".*

*Line 569: Replace "Air masses" by "The air masses".*

*Line 570: Replace "wherein air" by "while the air".*

*Line 572: Replace "of the trajectories" by "of trajectories".*

*Line 575: Replace "of air" by "of the air" and replace "on PMF" by "on the PMF".*

*Line 581: Replace "of sources" by "of the sources".*

*Line 587: Replace "shares" by "the shares".*

*Line 588: Replace "to elemental" by "to the elemental".*

*Line 602: Replace "at the ground" by "at ground".*

*Line 607: Replace "a.s.l. Half-hourly" by "a.s.l.. The half-hourly".*

*Line 610: Replace "where wind" by "where the wind".*

*Line 645: Replace "by the means" by "by means".*

*Line 655: Replace "different" by "the different".*

*Line 660: Replace "exploiting" by "exploitation".*

*Line 662: Replace "and Cl-rich" by "and the Cl-rich".*

*Line 666: Replace "and mixed" by "and a mixed".*

*Line 668: Replace "and traffic" by "and a traffic".*

*Line 670: Replace "showed" by "showed a" and replace "from south" by "and a traffic".*

*Line 671: Replace "while more" by "while a more" and replace "of identified" by "of the identified".*

*Line 676: Replace "showed strong" by "showed a strong" and replace "of soil" by "of the soil".*

All the above corrections have been made.

*Line 251, further within the manuscript, and Table 1: Many numeric data are given with too many significant figures; 2 significant figures suffice and 3 suffice in case the first significant figure is "1".*

The manuscript presents the concentrations of the PM-bound elements with a wide range of the levels (from below 1 up to more than 1000). In our opinion, the rounding of the concentrations, in particular the lower ones, could result in the loss of important information and would distort the meaning of the results that can be cited in further articles. In the whole manuscript, we consistently show the values of the concentrations with 1 decimal place.

---

## Author Response (AR3)

**Manuscript Number: acp-2021-253**

**Authors: Magdalena Reizer, Giulia Calzolai, Katarzyna Maciejewska, José A. G. Orza, Luca Carraresi, Franco Lucarelli, Katarzyna Juda-Rezler**

**Title: Measurement report: Receptor modelling for source identification of urban fine and coarse particulate matter using hourly elemental composition**

We would like to thank the Editor for very careful review of our paper, and for the comments. We took into account the comments in the revised version of the manuscript.

**Editor Decision: Publish subject to minor revisions (review by editor) (10 Aug 2021)**

*The following alterations, two of which I forgot to indicate in my previous report, are needed for the Main text before the manuscript can be published in ACP:*

*Line 151: Replace "use of Sunset" by "use of a Sunset".*

The correction has been made.

*Line 152: Replace "with flame" by "with a flame".*

The correction has been made.

*Line 251, further within the manuscript, and Table 1: Many numeric data are given with too many significant figures; 2 significant figures suffice and 3 suffice in case the first significant figure is "1". The authors seem to have misunderstood this comment. The first significant digit is the digit that differs from 0 when reading from left to right. Thus in Table 1, for Cr "1.3" and "1.1" should not be given with fewer significant digits; they could actually be extended with another significant digit. For Cl, most data are given with too many significant digits; for example, "1 682.5" should be replaced by "1 680". Incidentally, some data, e.g., for As, are given with only one significant digit, which does not suffice.*

Dear Editor, thank you for this comment, which we fully understood. However, we would like to keep current formatting of the numbers (with one exception). The reasons are the following: when preparing the manuscript, we checked the general rules of the mathematical notation in the ACP journal and there was nothing on the significant figures. On the other hand, the *SI Brochure: The International System of Units* – which is recommended by the ACP – states that "for numbers in a table, the format used should not vary within one column". We decided to apply this rule, using the consistent format of the concentration values in the whole manuscript, i.e. providing all values with one decimal place. This rule was not met in a few cases of very low concentrations (e.g. 0.05), therefore to be fully consistent in the formatting these values have been changed in Table 1 into "<0.1". We believe that this way of the formatting of the numbers can be easily understood by the readers and enables a direct comparison of the various orders of magnitude of the concentrations in Table 1. (The notation proposed by You will not meet the rule from SI Brochure, for example, after conversion we could have the values of 18.9, 25 and 0.63).

*Line 426: Replace "notable contribution" by "notable contributor".*

The correction has been made.